# FLOW MATCHING ACHIEVES ALMOST MINIMAX OPTIMAL CONVERGENCE

**Kenji Fukumizu**
The Institute of Statistical Mathematics/Preferred Networks
Tokyo, Japan
fukumizu@ism.ac.jp

**Taiji Suzuki**
University of Tokyo/RIKEN AIP
Tokyo, Japan
taiji@mist.i.u-tokyo.ac.jp

**Noboru Isobe**
University of Tokyo
Tokyo, Japan
nobo0409@g.ecc.u-tokyo.ac.jp

**Kazusato Oko**
University of Tokyo/RIKEN AIP
Tokyo, Japan
oko-kazusato@g.ecc.u-tokyo.ac.jp

**Masanori Koyama**
Preferred Networks/University of Tokyo
Tokyo, Japan
masanori.koyama@weblab.t-tokyo.ac.jp

## ABSTRACT

Flow matching (FM) has gained significant attention as a simulation-free generative model. Unlike diffusion models, which are based on stochastic differential equations, FM employs a simpler approach by solving an ordinary differential equation with an initial condition from a normal distribution, thus streamlining the sample generation process. This paper discusses the convergence properties of FM for large sample size under the $p$-Wasserstein distance. We establish that FM can achieve an almost minimax optimal convergence rate for $1 \le p \le 2$, presenting the first theoretical evidence that FM can reach convergence rates comparable to those of diffusion models. Our analysis extends existing frameworks by examining a broader class of mean and variance functions for the vector fields and identifies specific conditions necessary to attain almost optimal rates.

## 1 INTRODUCTION

Flow matching (FM) (Lipman et al., 2023; Albergo and Vanden-Eijnden, 2023; Liu et al., 2023b) is a recent simulation-free generative model that produces samples of the target distribution by solving an ordinary differential equation (ODE) initialized with a source normal distribution. The vector field to define the ODE is trained by neural networks with the teaching data of random conditional vectors. This approach bypasses the computationally intensive Monte Carlo sampling required in the diffusion model, which is currently the standard in generative modeling. Various variations have been proposed to refine the learning of vector fields, such as OT-CFM (Tong et al., 2024), rectified flow (Liu et al., 2023b), consistent velocity field (Yang et al., 2024), equivariant flow (Klein et al., 2023), etc. A series of studies also emerge from the viewpoint of interpolating distributions (Albergo et al., 2023c;a).

FM has already been applied to various domains with promising performance. Among many others, the rectified flow method has been extended to high-resolution text image generation (Esser et al., 2024), and there are also many works on the application of FM to molecule generation (Hoogeboom et al., 2022; Guan et al., 2023; Bose et al., 2023; Dunn and Koes, 2024), text generation (Hu et al., 2024), speech generation (Le et al., 2023), motion synthesis (Hu et al., 2023), etc.

Although the methods have been developed on the solid theoretical basis of the flows and continuity equation, their statistical behaviors remain less understood. Recent works have established the convergence of the FM estimator to the true distribution under some distributional metrics (Albergo and Vanden-Eijnden, 2023; Benton et al., 2023b). Beyond the convergence, more detailed understandings,

such as convergence rates, are still an open question. In contrast, diffusion models have gained various theoretical understandings, including the convergence rate in terms of the number of steps (Chen et al., 2023; Benton et al., 2023a) and the sample size (Oko et al., 2023; Zhang et al., 2024). Among others, Oko et al. (2023) has shown that diffusion models achieve the minimax optimal convergence rate for a large sample size under the total variation metric and the almost minimax optimal rate under the 1-Wasserstein distance, where the max is taken over the true densities of the Besov space. This result theoretically supports the high generation ability of diffusion models.

This paper aims to bridge this gap by demonstrating that FM can achieve an almost minimax optimal convergence rate for a large sample size under the $p$-Wasserstein distance $W_p$ for $1 \leq p \leq 2$, suggesting that FM has a theoretical ability comparable to diffusion models. This problem is significant for comparing the ability of FM methods and diffusion models, and revealing the difference between SDE and ODE in the generative models. Drawing on the methodologies of Oko et al. (2023), our analysis not only extends to a broader class of mean and variance parameters of Gaussian smoothing for conditional vector fields, but also specifies the conditions on these parameters under which the almost minimax optimal convergence rate can be achieved.

The contributions of this paper are as follows.

- We establish that a widely used class of conditional FM methods achieves an almost minimax optimal convergence rate under the $p$-Wasserstein distance ($1 \leq p \leq 2$), marking the first theoretical demonstration of such optimal performance of FM.
- We provide an analytical derivation of the convergence rate under various settings of the parameters, mean and variance, to make a path that connects a source and target point.
- We reveal that the variance parameter, which specifies the contribution of the source, must be decreased around the target at a specific rate to attain an almost minimax optimal convergence rate.

## 2 FLOW MATCHING

Throughout the paper, data are in the $d$-dimensional space $\mathbb{R}^d$. The $d$-dimensional normal distribution with mean $\boldsymbol{\mu}$ and covariance matrix $V$ is denoted by $N_d(\boldsymbol{\mu}, V)$. For a probability $P_a$ with index $a$, the lowercase $p_a$ denotes its probability density function (p.d.f.).

### 2.1 REVIEW OF FLOW MATCHING

This subsection provides a general review of FM, following Lipman et al. (2023) and Tong et al. (2024). The aim of FM is to generate samples from the true probability $P_{true}$. FM methods realize it by a flow $\boldsymbol{\varphi}_{[\tau]}(\boldsymbol{x})$ ($\tau \in [0, 1]$)[1] that maps a sample from the standard normal distribution $N_d(0, I_d)$ to that of $P_{true}$. The flow $\boldsymbol{\varphi}_{[\tau]}(\boldsymbol{x})$ is defined by a solution to the ODE

$$\frac{d}{d\tau}\boldsymbol{x}_{[\tau]} = \boldsymbol{v}_{[\tau]}(\boldsymbol{x}_{[\tau]}) \qquad (\tau \in [0, 1])$$

given by a desired vector field $\boldsymbol{v}_{[\tau]}$. FM generates a sample by solving the ODE with an initial point $\boldsymbol{x}_{[0]}$ from $P_{[0]} = N_d(0, I_d)$; in other words, the distribution at time $\tau$ is the pushforward $P_{[\tau]} = \boldsymbol{\varphi}_{[\tau]\#}P_{[0]}$. The pushforward $P_{[1]}$ is expected to approximate $P_{true}$. In practice, we need to construct the vector field given training data $\{\boldsymbol{x}^i\}_{i=1}^n$ of size $n$, which is i.i.d. samples from $P_{true}$.

The relation between the vector field $\boldsymbol{v}_{[\tau]}(\boldsymbol{x})$ and the p.d.f. $p_{[\tau]}(\boldsymbol{x})$ is given by the *continuity equation*:

$$\frac{\partial}{\partial\tau}p_{[\tau]}(\boldsymbol{x}) + \text{div}\big(p_{[\tau]}(\boldsymbol{x})\boldsymbol{v}_{[\tau]}(\boldsymbol{x})\big) = 0.$$

Typically, a neural network (NN) is used to construct $\boldsymbol{v}_{[\tau]}(\boldsymbol{x})$. However, it is not obvious how to prepare the desired $\boldsymbol{v}_{[\tau]}(\boldsymbol{x})$ to teach NN. In FM methods, conditional random vectors $\boldsymbol{v}_{[\tau]}(\boldsymbol{x}_{[\tau]}|\boldsymbol{z})$ given $\boldsymbol{z}$, which are to be easily prepared, are used to teach NN; a location $\boldsymbol{x}_{[\tau]}$ is sampled by a conditional probability $P_{[\tau]}(\boldsymbol{x}_{[\tau]}|\boldsymbol{z})$ and the vector $\boldsymbol{v}_{[\tau]}(\boldsymbol{x}_{[\tau]}|\boldsymbol{z})$ is assigned at $\boldsymbol{x}_{[\tau]}$ as teaching data.

---

[1] We use $[\tau]$ to denote the time $\tau \in [0, 1]$ in this section and preserve $\boldsymbol{x}_t$ for the reverse time indexing, which is adopted from Section 4 to align with the notation of diffusion models.

Throughout this paper, the condition is given by $z = (x_{[0]}, x_{[1]})$ with $x_{[0]} \sim P_{[0]}$ and $x_{[1]} \sim P_{true}$. The vector $v_{[\tau]}(x_{[\tau]}|z)$ is made so that it satisfies the conditional continuity equation:

$$\frac{\partial}{\partial \tau} p_{[\tau]}(x|z) + \text{div}\left(p_{[\tau]}(x|z)v_{[\tau]}(x|z)\right) = 0. \tag{1}$$

A typical construction of $x_{[\tau]}$ is to use a path $x_{[\tau]}$ ($\tau \in [0,1]$) from $x_{[0]}$ to $x_{[1]}$ and define the conditional vector by its time derivative $v_{[\tau]}(x|z) := \frac{d}{d\tau}x_{[\tau]}$ (see Sec. 2.2). For a deterministic path, $P_{[\tau]}(x_{[\tau]}|z)$ is the delta function at a point in the path $x_{[\tau]}$.

Note that, given $(x, \tau)$, the vector $v_{[\tau]}(x|z)$ is random by the choice of $z = (x_{[0]}, x_{[1]})$; different vectors may be assigned to the same location $(x, \tau)$. Most importantly, by averaging over $z \sim Q$, where $Q$ is the joint distribution with marginals $P_{[0]}$ and $P_{[1]}$, we can see that the p.d.f. of $x$ at time $\tau$,

$$p_{[\tau]}(x) = \int p_{[\tau]}(x|z) dQ(z), \tag{2}$$

and the averaged vector field $v_{[\tau]}(x)$ given by

$$v_{[\tau]}(x) := \int v_{[\tau]}(x|z) p_{[\tau]}(z|x) dz, \qquad p_{[\tau]}(z|x) := \frac{p_{[\tau]}(x|z)q(z)}{p_t(x)} \tag{3}$$

satisfy the continuity equation

$$\frac{\partial}{\partial \tau} p_{[\tau]}(x) + \text{div}\left(p_{[\tau]}(x)v_{[\tau]}(x)\right) = 0. \tag{4}$$

This provides the theoretical basis for FM methods; the averaged vector field $v_{[\tau]}$ transports $N_d(0, I_d)$ to $P_{true}$. $v_{[\tau]}$ is learned by a NN $\phi(x, \tau)$ with noisy training data $\{(x_{[\tau]}, \tau), v_{[\tau]}(x_{[\tau]}|z)\}$. Empirically, the conditional $v_{[\tau]}(x|z)$ is given by the random sample $z = (x_{[0]}, x^i)$ and the location $x_{[\tau]} \sim P_{[\tau]}(x|z)$ (or path) with uniform $\tau$. The NN is trained with the mean square error (MSE):

$$\min_\phi \mathbb{E}\|\phi(x_{[\tau]}, \tau) - v_{[\tau]}(x_{[\tau]}|z)\|^2. \tag{5}$$

Note that $\phi(x_{[\tau]}, \tau)$ does not depend on $z$. Since the MSE minimizer is the conditional expectation of the teaching data, the empirical minimizer $\widehat{\phi}$ is an estimator of $v_{[\tau]}(x)$. Using the estimator $\widehat{\phi}$ and the corresponding flow $\widehat{\varphi}_{[\tau]}$ given by ODE, we obtain the estimator $\widehat{P}_{[1]}$ for $P_{true}$ by sampling. In practice, to reduce the variance of $(x_{[0]}, x_{[1]})$ and simplify the ODE solution, the optimal transport for pairing $x_{[0]}$ and $x_{[1]}$ is applied effectively (Tong et al., 2024; Pooladian et al., 2023).

## 2.2 PATH CONSTRUCTION

This paper focuses on the following class of paths to construct the conditional vector field. Let $x_{[0]} \sim P_{[0]} = N_d(0, I_d)$, and $x_{[1]}$ be a sample of $P_{[1]}$ (or the empirical distribution $\hat{P}_{train} = (1/n)\sum_{i=1}^n \delta_{x^i}$ in practice). A conditional path is defined by

$$x_{[\tau]} := \sigma_{[\tau]} x_{[0]} + m_{[\tau]} x_{[1]} \qquad (0 \le \tau \le 1, \sigma_{[\tau]} > 0, m_{[\tau]} > 0). \tag{6}$$

We assume that $\sigma_t$ and $m_t$ are monotonic, $\sigma_{[\tau]} \to 1, m_{[\tau]} \to 0$ as $\tau \to 0^+$, and $\sigma_{[\tau]} \to 0, m_{[\tau]} \to 1$ as $\tau \to 1^-$. Let $\sigma'_{[\tau]}$ ($m'_{[\tau]}$, resp.) be the time derivative of $\sigma_{[\tau]}$ ($m_{[\tau]}$, resp.). With sampling $\tau \sim \text{Unif}[0,1]$, a random conditional vector is assigned at $(x_{[\tau]}, \tau) \in \mathbb{R}^d \times [0,1]$ by

$$v_{[\tau]}(x_{[\tau]}|x_{[0]}, x_{[1]}) := \sigma'_{[\tau]} x_{[0]} + m'_{[\tau]} x_{[1]}. \tag{7}$$

Note that, due to $x_{[0]} \sim N_d(0, I_d)$, the distribution of $x_{[\tau]}$ given $x_{[1]}$ equals $P_{[\tau]}(x_{[\tau]}|x_{[1]}) = N_d(m_{[\tau]} x_{[1]}, \sigma_{[\tau]}^2 I_d)$, and thus we call $m_{[\tau]}$ and $\sigma_{[\tau]}^2$ the mean and variance parameters, respectively. Since (6) leads $x_{[0]} = (x_{[\tau]} - m_t x_{[1]})/\sigma_{[\tau]}$, the conditional vector (7) is written as

$$v_{[\tau]}(x_{[\tau]}|x_{[1]}) = \sigma'_{[\tau]} \frac{x_{[\tau]} - m_t x_{[1]}}{\sigma_{[\tau]}} + m'_{[\tau]} x_{[1]}. \tag{8}$$

This class covers some popular constructions of conditional vector fields in the literature.

- **Affine path:** one of the most popular constructions is the following,

$$\boldsymbol{x}_{[\tau]} := (1-\tau)\boldsymbol{x}_{[0]} + \tau\boldsymbol{x}_{[1]}, \qquad \boldsymbol{v}_{[\tau]}(\boldsymbol{x}_{[\tau]}|\boldsymbol{x}_{[1]}) = \boldsymbol{x}_{[1]} - \boldsymbol{x}_{[0]}.$$

This corresponds to $m_{[\tau]} = \tau$ and $\sigma_{[\tau]} = 1-\tau$. In Lipman et al. (2023) $\boldsymbol{x}_{[0]}$ and $\boldsymbol{x}_{[1]}$ are generated independently, while in Tong et al. (2024) they are taken by the optimal transport in a minibatch. This case is covered by our result, which does not depend on the construction of joint distribution.

- **Diffusion:** Lipman et al. (2023) presents the diffusion path, which corresponds to the deterministic probability flow (Song et al., 2020). The conditional density is given by $p_{[\tau]}(\boldsymbol{x}_{[\tau]}|\boldsymbol{x}_{[1]} = \boldsymbol{y}) = N_d(m_{[\tau]}\boldsymbol{y}, \sigma_{[\tau]}^2 I_d)$. The setting $\sigma_{[\tau]}^2 = 1 - m_{[\tau]}^2$ and $\sigma_{[\tau]} \sim \sqrt{1-\tau}$ is typically used.

## 3 CONVERGENCE RATE OF FLOW MATCHING

We assume that the true density $p_{[1]}$ is included in the *Besov space* $B_{p',q'}^s$ ($s > 0, 0 < p', q' \leq \infty$) on the unit cube $[-1,1]^d$, while the results can be extended to any size straightforwardly. The parameter $s$ specifies the degree of smoothness and is most relevant in this paper. The definition of the Besov space is deferred to Appendix A.1. We use the $r$-Wasserstein distance $W_r$ to measure the accuracy of the estimator. The distance $W_r$ of the probabilities $P_1$ and $P_2$ on $\mathbb{R}^d$ is defined by

$$W_r(P_1, P_2) := \left(\inf_{Q \in \Gamma(P_1, P_2)} \int \|\boldsymbol{x}_1 - \boldsymbol{x}_2\|^r dQ(\boldsymbol{x}_1, \boldsymbol{x}_2)\right)^{1/r}, \tag{9}$$

where $\Gamma(P_1, P_2)$ denotes the joint distribution of $(\boldsymbol{x}_1, \boldsymbol{x}_2)$ with marginals $P_1$ and $P_2$. It is well known that $W_r(P_1, P_2) \leq W_{r'}(P_1, P_2)$ holds for $r' \geq r \geq 1$.

As discussed in Sec. 3.1, to obtain an accurate estimator, we need to adopt early stopping of ODE and use $\hat{P}_{[1-T_0]}$ with small $T_0$. Our aim is to derive a bound of $W_p(\hat{P}_{[1-T_0]}, P_{true})$ for a large sample $n \to \infty$. The informal version of our main result is summarized in the following theorem.

**Theorem 1** (Informal). *Suppose that the target probability $P_{[1]}$ has p.d.f. $p_{[1]}$ in the Besov space $B_{p',q'}^s([-1,1]^d)$ of smoothness degree $s$, and that $n$ training data $\{x^{(i)}\}_{i=1}^n$ is i.i.d. samples from $P_{[1]}$. Assume that $\sigma_{[\tau]} \sim (1-\tau)^\kappa$ ($\tau \to 1^-$) with $\kappa \geq 1/2$, the conditinal vector field is given by* (6) *and* (7)*, and that time-divided neural networks are used (see Sec. 4.3). Then, under several assumptions, the FM estimator $\hat{P}_{[1-T_0]}$ with $T_0 = n^{-R_0}$ with appropriate $R_0$ satisfies, for any $\delta > 0$,*

$$\mathbb{E}[W_2(\hat{P}_{[1-T_0]}, P_{true})] = O\left(n^{-\frac{s+(2\kappa)^{-1}-\delta}{2s+d}}\right) \qquad (n \to \infty), \tag{10}$$

*where $\mathbb{E}$ denotes the expectation over the training data.*

It is known that a lower bound of the minimax convergence rate exists for the Wasserstein distance for probability estimation. We use the notation $\gtrsim$ to mean the lower bound up to a constant factor.

**Proposition 2** (Niles-Weed and Berthet (2022)). *Let $p', q' \geq 1$, $s > 0$, $r \geq 1$, and $d \geq 2$. Then,*

$$\inf_{\hat{P}} \sup_{p \in B_{p',q'}^s([-1,1]^d)} \mathbb{E}[W_r(\hat{P}, P)] \gtrsim n^{-\frac{s+1}{2s+d}} \qquad (n \to \infty),$$

*where $\hat{P}$ runs over all estimators based on $n$ i.i.d. samples from $P$.*

For $\kappa = 1/2$, by Theorem 1 and Proposition 2, the upper bound $n^{-\frac{s+1-\delta}{2s+d}}$ is almost the optimal convergence rate up to an arbitrarily small $\delta > 0$. In addition, this convergence rate coincides with that of the diffusion model given in Oko et al. (2023) for $W_1$. The above result indicates that the flow matching is as good as the diffusion model regarding the minimax convergence rate under $W_1$, where the max in minimax means sup over the Besov space.

### 3.1 KERNEL DENSITY ESTIMATION AND EARLY STOPPING OF ODE

In practice, with conditional density $P_{[\tau]}(\boldsymbol{x}|\boldsymbol{x}_{[1]}) = N_d(m_{[\tau]}\boldsymbol{x}_{[1]}, \sigma_{[\tau]}^2 I_d)$, the parameter $\sigma_{[1]}$ is often set as a small positive value $\sigma_{[1]} = \sigma_{min} > 0$ so that (7) is well defined up to $\tau = 1$ (e.g. Lipman et al., 2023). If $\boldsymbol{x}_{[1]}$ is sampled from $\hat{P}_{train} = \frac{1}{n}\sum_{j=1}^n \delta_{\boldsymbol{x}^j}$, the obtained distribution equals to

$$\hat{p}_{[1]}(\boldsymbol{x}) = \int p_{[1]}(\boldsymbol{x}|\boldsymbol{x}_{[1]})d\hat{P}_{train}(\boldsymbol{x}_{[1]}) = \frac{1}{n}\sum_{j=1}^n \frac{1}{(2\pi\sigma_{min}^2)^{d/2}} \exp\left(-\frac{\|\boldsymbol{x}-\boldsymbol{x}^j\|^2}{2\sigma_{min}^2}\right),$$

which is exactly the kernel density estimator (KDE) with the Gaussian kernel of bandwidth $\sigma_{\min}$. If the ODE is solved up to $\tau = 1$ rigorously, the pushforward realizes this KDE. As is well known (Scott, 1992), the convergence rate of this KDE under MSE is $O(n^{-4/(4+d)})$ at best by choosing the optimal $\sigma_{\min}$ depending on $n$, which is much slower than the optimal rate $n^{-2s/(2s+d)}$ under MSE for the true density in $B^s_{p',q'}(I^d)$ (Liu et al., 2023a). Based on this consideration, we discuss the early stopping of the ODE, where we stop at $\tau = 1 - T_0$ with small $T_0 > 0$ and consider the convergence rate of the estimator $\widehat{p}_{[1-T_0]}$. Notice that $\widehat{p}_{[1-T_0]}$ differs from KDE, since it is given by the trained vector field. For diffusion models, Oko et al. (2023) and Zhang et al. (2024) also discuss the estimator obtained by stopping the reverse SDE at $T_0 > 0$ to derive the convergence rate.

## 3.2 RELATED WORKS

Among many literatures on the statistical convergence of diffusion models, the most relevant to this work is Oko et al. (2023). Although our analysis is based on Oko et al. (2023) and derives comparable results, there are significant differences. First, we analyze the more general settings for $m_{[\tau]}$ and $\sigma_{[\tau]}$ in the conditional distribution $P_{[\tau]}(\boldsymbol{x}|\boldsymbol{y}) = N_d(m_{[\tau]}\boldsymbol{y}, \sigma^2_{[\tau]}I)$; Oko et al. (2023) considers only the case of $\sigma_t \sim \sqrt{t}$ and $m_t \sim 1 - t$ (in reverse time $t$), which is a typical choice for diffusion models. Consequently, we have shown that for $\sigma_{[\tau]} \sim (1 - \tau)^\kappa$ with $\kappa \geq 1/2$, only $\kappa = 1/2$ achieves the almost minimax optimal convergence rate. Second, due to the difference between the ODE and diffusion processes, the proof technique for relating the Wasserstein metric and the $L_2$-risk is very different. Our technique is based on Alekseev-Gröbner lemma to derive the bound for $r$-Wasserstein with $1 \leq r \leq 2$, while Oko et al. (2023) obtained the bound only for 1-Wasserstein. Third, this is the first theoretical result for FM showing a convergence rate that is almost optimal. Although FM has been recently used in many applications with competitive results to diffusion models, theoretical comparisons in terms of convergence rates have been lacking. The results of this paper show that both FM and DM can attain the same almost minimax optimal convergence rate for generalization error.

For FM, there are some recent works on convergence. Albergo and Vanden-Eijnden (2023) and Benton et al. (2023b) relate the Wasserstein distance to the $L_2$-risk of the vector fields and show convergence for a large sample size, but did not derive a convergence rate. Jiao et al. (2024) discusses convergence rates of FM applied in the latent space of the autoencoder and considers the discretization effect of the numerical ODE solution in their analysis. However, they did not include the degree of smoothness in developing the convergence rate. Albergo et al. (2023b) present a unifying view of the theory of diffusion models and FM with the upper bounds of discrepancy measures.

## 4 THEORETICAL DETAILS

This section rigorously presents the main result with the assumptions and shows the proof outline. In the sequel, we use *reverse time index* $t = 1 - \tau$ ($\tau \in [0, 1]$); $t = 0$ for $P_{true}$ and $t = 1$ for $N_d(0, I_d)$, which align with the notations of the diffusion models. We use $\mathrm{poly}(\log n)$ to indicate the term of $O(\log^r n)$-order with some natural number $r$, and $\tilde{O}(n^\alpha)$ to mean the order up to $\mathrm{poly}(\log n)$ factor.

## 4.1 PROBLEM SETTING AND ASSUMPTIONS

With reverse time $t$, the definitions (7), (2), and (3) are modified by replacing $[1 - t]$ with $t$;

$$P_t = P_{[1-t]}, \qquad P_0 = P_{true}, \qquad P_1 = N_d(0, I_d).$$

The flows $\boldsymbol{\varphi}_t$ and $\widehat{\boldsymbol{\varphi}}_t$ are defined by solving the ODE from $t = 1$ in the reverse time direction:

$$\frac{d}{dt}\boldsymbol{\varphi}_t(\boldsymbol{x}) = \boldsymbol{v}_t(\boldsymbol{\varphi}_t(\boldsymbol{x})), \qquad \frac{d}{dt}\widehat{\boldsymbol{\varphi}}_t(\boldsymbol{x}) = \widehat{\boldsymbol{v}}_t(\boldsymbol{\varphi}_t(\boldsymbol{x})), \tag{11}$$

where $\boldsymbol{v}_t(\boldsymbol{x})$ and $\widehat{\boldsymbol{v}}_t(\boldsymbol{x})$ are the vector field (3) and its neural estimate, respectively. The distributions at $t \in [0, 1]$ are given by

$$P_t = (\boldsymbol{\varphi}_t)_\# P_1, \qquad \widehat{P}_t = (\widehat{\boldsymbol{\varphi}}_t)_\# P_1, \tag{12}$$

where $(\boldsymbol{\varphi}_t)_\#$ and $(\widehat{\boldsymbol{\varphi}}_t)_\#$ denote the pushforward by the respective flows $\boldsymbol{\varphi}_t$ and $\widehat{\boldsymbol{\varphi}}_t$.

In the remainder of this paper, $\delta > 0$ is an arbitrarily small positive value. As in Oko et al. (2023), we introduce $N$ to specify the number of basis functions of the $B$-spline for approximating $p_t(\boldsymbol{x})$

and $\boldsymbol{v}_t(\boldsymbol{x})$. This number $N$ depends on the sample size $n$ ($N = n^{\frac{d}{2s+d}}$ is used), balancing the approximation error and complexity of the $B$-spline and NN. We set the stopping time $T_0 = N^{-R_0}$ as discussed in Sec. 3.1 ($R_0$ is specified later), and solve the ODE from 1 to $T_0$. For simplicity, the $d$ dimensional cube $[-1,1]^d$ and the reduced cube $[-1+N^{-(1-\kappa\delta)}, 1-N^{-(1-\kappa\delta)}]^d$ are denoted by $I^d$ and $I_N^d$, respectively, where $\kappa > 0$ is specified below in (A3). We make the following assumptions.

(A1) The target probability $P_0$ has support $I^d$ and its p.d.f. $p_0$ satisfies $p_0 \in B_{p',q'}^s(I^d)$ and $p_0 \in B_{p',q'}^{\check{s}}(I^d \backslash I_N^d)$ with $\check{s} > \max\{6s-1, 1\}$.

(A2) There exists $C_0 > 0$ such that $C_0^{-1} \leq p_0(\boldsymbol{x}) \leq C_0$ for all $\boldsymbol{x} \in I^d$.

(A3) There is $\kappa \geq 1/2$, $b_0 > 0$, $\tilde{\kappa} > 0$, and $\tilde{b}_0 > 0$ such that

$$\sigma_t = b_0 t^\kappa, \qquad 1 - m_t = \tilde{b}_0 t^{\tilde{\kappa}}$$

for sufficiently small $t \geq T_0$. Also, there are $D_0 > 0$ and $K_0 > 0$ such that

$$D_0^{-1} \leq \sigma_t^2 + m_t^2 \leq D_0, \qquad |\sigma_t'| + |m_t'| \leq N^{K_0} \quad (\forall t \in [T_0, 1]).$$

(A4) If $\kappa = 1/2$, there is $b_1 > 0$ and $D_1 > 0$ such that for any $0 \leq \gamma < R_0$

$$\int_{T_0}^{N^{-\gamma}} \{(\sigma_t')^2 + (m_t')^2\} dt \leq D_1 (\log N)^{b_1}.$$

(A5) There is a constant $C_L > 0$ such that $\|\frac{\partial}{\partial \boldsymbol{x}} \int \boldsymbol{y} p_t(\boldsymbol{y}|\boldsymbol{x}) d\boldsymbol{y}\|_{op} \leq C_L$ for any $t \in [T_0, 1]$.

The higher degree of smoothness is assumed around the boundary of $I^d$ in (A1) for a technical reason to compensate for the nondifferentiability of $p_0(x)$ at the boundary by (A2). In (A3), it may be more natural to require $\sigma_t^2 + m_t^2 = 1$ so that signal power can be maintained. However, in this paper, to pursue the flexibility of choosing $\sigma_t$ and $m_t$, we allow bounded changes of $\sigma_t^2 + m_t^2$ over $t$. (A4) is required to limit the complexity of the neural network model (see Lemma 5). In (A3), $\kappa$ is assumed to be not less than $1/2$, because for $\kappa < 1/2$, the integral $\int_{T_0}(\sigma_t')^2 dt$ with $T_0 = N^{-R_0}$ diverges to infinity as $N \to +\infty$, which causes the divergence of the complexity bound in Lemma 5. Note that the boundary case $\kappa = 1/2$ is, in fact, popularly used for the diffusion model. In this case, $(\sigma_t')^2$ is the order $1/t$ for $t \to 0^+$ and the integral from $T_0$ is of the order $\log N$, which still diverges to infinity as $n \to \infty$. As discussed in Section 4.3, we consider this integral only for a short time interval, and we will see that the $W_2$ distance converges to zero as $n \to \infty$. (A5) is made to bound the Lipschitz factor in Theorem 3 under (A3) (see Lemma 10).

## 4.2 Generalization bound

It is known (Albergo and Vanden-Eijnden, 2023; Benton et al., 2023b) that, given two vector fields, the $W_2$-distance of the pushforwards of the same distribution by the corresponding flows admits an upper bound by the $L_2$-risk of the vector fields;

**Theorem 3.** *Let $\boldsymbol{v}_t(\boldsymbol{x})$ and $\widehat{\boldsymbol{v}}_t(\boldsymbol{x})$ be vector fields such that $\boldsymbol{x} \mapsto \widehat{\boldsymbol{v}}_t(\boldsymbol{x})$ is $L_t$-Lipschitz for each $t$, and $P_t$ and $\widehat{P}_t$ be the pushfowards of distribution $P_0$ by the corresponding flows at time $t$ from $t = 0$. Then, for any $t \in [0,1]$, we have*

$$W_2(\widehat{P}_t, P_t) \leq \sqrt{t} \left( \int_0^t \int e^{2\int_s^t e^{L_u} du} \|\widehat{\boldsymbol{v}}_s(\boldsymbol{x}) - \boldsymbol{v}_s(\boldsymbol{x})\|^2 dP_s(\boldsymbol{x}) d\boldsymbol{x} ds \right)^{1/2}. \tag{13}$$

See Appendix B for the proof. From Theorem 3, we can consider the $L_2$-error $\mathbb{E}[\int \int \|\widehat{\boldsymbol{v}}(\boldsymbol{x}, s) - \boldsymbol{v}(\boldsymbol{x}, s)\|^2 dP_s(\boldsymbol{x}) d\boldsymbol{x} ds]$ of the vector field to obtain the bound of the $W_2$ distance of the distributions. From the fact $W_r \leq W_{r'}$ ($1 \leq r \leq r'$), the same upper bound holds for $W_r$ for $1 \leq r \leq 2$.

We first review a general method for bounding the generalization. We consider training within the general time interval $[T_\ell, T_u]$ where $T_0 \leq T_\ell < T_u \leq 1$. For an estimator $\boldsymbol{\phi}(\boldsymbol{x}, t)$ of the true vector field $\boldsymbol{v}_t(\boldsymbol{x})$, we define the loss function $\ell_{\boldsymbol{\phi}}^{T_\ell, T_u}(\boldsymbol{x})$ for $x \in I^d$ by

$$\ell_{\boldsymbol{\phi}}^{T_\ell, T_u}(\boldsymbol{x}) := \int_{T_\ell}^{T_u} \int \|\boldsymbol{\phi}(\boldsymbol{x}_t, t) - \boldsymbol{v}_t(\boldsymbol{x}_t|\boldsymbol{x})\|^2 p_t(\boldsymbol{x}_t|\boldsymbol{x}) d\boldsymbol{x}_t dt, \tag{14}$$

where $\boldsymbol{x}$ is the condition of $\boldsymbol{v}_t(\boldsymbol{x}_t|\boldsymbol{x})$. Although the definition depends on $T_\ell$ and $T_u$, we omit them when there is no confusion. Given the training data $\{\boldsymbol{x}^i\}_{i=1}^n$, the vector field is trained with the teaching data $\boldsymbol{v}_t(\boldsymbol{x}_t|\boldsymbol{x}^i)$ at the location $(\boldsymbol{x}_t, t)$ ($t \in [T_\ell, T_u]$), which is sampled from $p_t(\boldsymbol{x}_t|\boldsymbol{x}^i)$ and the uniform distribution $U([T_\ell, T_u])$. Note that given $\boldsymbol{x}^i$, we can generate any number of $(\boldsymbol{x}_t, t)$. Thus, the sampling error in (14) is negligible and the training by a NN can be regarded as minimization of

$$\frac{1}{n}\sum_{i=1}^n \ell_{\boldsymbol{\phi}}(\boldsymbol{x}^{(i)}). \tag{15}$$

See Oko et al. (2023, Section 4) for the discussion of the effect of sampling.

Let $\widehat{\boldsymbol{\phi}}$ be the minimizer of (15) among the function class $\mathcal{S}$. The generalization error is then given by

$$\mathcal{E}_{gen} := \mathbb{E}\Big[\int\int\int_{T_\ell}^{T_u} \|\widehat{\boldsymbol{\phi}}(\boldsymbol{x}, t) - \boldsymbol{v}_t(\boldsymbol{x})\|^2 p_t(\boldsymbol{x}|\boldsymbol{y}) dt d\boldsymbol{x} p_0(\boldsymbol{y}) d\boldsymbol{y}\Big] = \mathbb{E}\Big[\int \ell_{\widehat{\boldsymbol{\phi}}}(\boldsymbol{y}) p_0(\boldsymbol{y}) d\boldsymbol{y}\Big]. \tag{16}$$

Let $\mathcal{L} := \{\ell_{\boldsymbol{\phi}} \mid \boldsymbol{\phi} \in \mathcal{S}\}$ and $\mathcal{N}(\mathcal{L}, \|\cdot\|_{L^\infty(I^d)}, \varepsilon)$ be the covering number of the function class $\mathcal{L}$ with the $\|\cdot\|_{L^\infty(I^d)}$-norm. Then, a standard argument on the generalization error analysis derives the following upper bound (see Oko et al. (Theorem C.4, 2023) and also Hayakawa and Suzuki (2020)).

**Theorem 4.** *The generalization error of the minimizer of* (15) *among* $\boldsymbol{\phi} \in \mathcal{S}$ *is upper bounded by*

$$\mathcal{E}_{gen} \le 2 \inf_{\boldsymbol{\phi} \in \mathcal{S}} \int\int_{T_\ell}^{T_u} \|\boldsymbol{\phi}(\boldsymbol{x}, t) - \boldsymbol{v}_t(\boldsymbol{x})\|^2 p_t(\boldsymbol{x}) dt d\boldsymbol{x}$$

$$+ \frac{\sup_{\boldsymbol{\phi} \in \mathcal{S}} \|\ell_{\boldsymbol{\phi}}\|_{L^\infty(I^d)}}{n}\left(\frac{37}{9}\log\mathcal{N}(\mathcal{L}, \|\cdot\|_{L^\infty(I^d)}, \varepsilon) + 32\right) + 3\varepsilon. \tag{17}$$

From Theorems 3 and 4, it suffices to consider the approximation error (1st term) and complexity (2nd term) in (17) for deriving the $W_2$ distributional bound.

### 4.2.1 COMPLEXITY TERM IN GENERALIZATION BOUND

We first consider the complexity term, where the class $\mathcal{S}$ is given by NN. A class of NN $\mathcal{M}(L, W, S, B)$ with height $L$, width $W$, sparsity constraint $S$, and norm constraint $B$ is defined as

$$\mathcal{M}(L, W, S, B) := \{\psi_{A^{(L)}, b^{(L)}} \circ \cdots \circ \psi_{A^{(2)}, b^{(2)}}(A^{(1)}\boldsymbol{x} + b^{(1)}) \mid A^{(i)} \in \mathbb{R}^{W_{i+1} \times W_i}, b^{(i)} \in \mathbb{R}^{W_{i+1}},$$

$$\sum_{i=1}^L (\|A^{(i)}\|_0 + \|b^{(i)}\|_0) \le S, \max_i \|A^{(i)}\|_\infty \vee \|b^{(i)}\|_\infty \le B\},$$

where $\psi_{A,b}(z) = A\,\mathrm{ReLU}(z) + b$. As shown in Theorems 7 and 8 later, it suffices to consider the NNs that satisfy

$$\|\boldsymbol{\phi}(\boldsymbol{x}, t)\|_\infty \le D\big(|\sigma_t'|\sqrt{\log n} + |m_t'|\big)$$

for some constant $D$. Also, we can see in Lemma A.2 that $\boldsymbol{x} \mapsto \boldsymbol{v}_t(\boldsymbol{x})$ is Lipschitz continuous with Lipschitz constant proportial to $1/t$ under (A3) and (A5). Reflecting these facts, we define the following NN class for training the vector field:

$$\mathcal{H}_n := \big\{\boldsymbol{\phi} \in \mathcal{M}(L, W, S, B) \mid \|\boldsymbol{\phi}(\cdot, t)\|_\infty \le D\big(|\sigma_t'|\sqrt{\log n} + |m_t'|\big) \text{ for } \forall t \in [T_0, 1],$$

$$\boldsymbol{x} \mapsto \boldsymbol{\phi}(\boldsymbol{x}, t) \text{ is } L_t\text{-Lipschitz for each } t \in [T_0, 1] \text{ where } L_t = \tilde{C}_L/t\big\}, \tag{18}$$

where $D$ and $\tilde{C}_L$ are some positive constants.

The supremum norm and the covering number in Theorem 4 are given in the following lemmas.

**Lemma 5.** *Let* $T_0 \le T_\ell < T_u \le 1$. *Under Assumption* (A4), *there is* $C_s > 0$ *such that*

$$\sup_{\boldsymbol{\phi} \in \mathcal{H}_n} \|\ell_{\boldsymbol{\phi}}\|_{L^\infty(I^d)} \le C_s (\log n)^{b+1}, \tag{19}$$

*where* $b = b_1$ *in* (A4) *for* $\kappa = 1/2$, *and* $b = 0$ *for* $\kappa > 1/2$.

See Appendix C.1 for the proof. To obtain this bound, we need to impose the upper bound of $\boldsymbol{\phi}$ as in (18). In practice, the vectors in the teaching data satisfy this upper bound, and thus $\boldsymbol{\phi}$ will naturally satisfy the same bound by the least square error solution. The following bound of the covering number for neural networks is given by Suzuki (Lemma 3, 2019).

**Lemma 6.** *For the function class* $\mathcal{H}_n$, *the covering number satisfies*

$$\log\mathcal{N}(\mathcal{L}, \|\cdot\|_{L^\infty(I^d)}, \varepsilon) \le SL \log\big(\varepsilon^{-1}\|W\|_\infty Bn\big).$$

### 4.2.2 Approximation error for small $t$

Recall that $N$ specifies the number of basis functions of the $B$-spline for the approximation. We derive upper bounds of the approximation error of the NN model $\mathcal{M}(L, W, S, B)$, where $L, W, S$, and $B$ are specified in terms of $N$. We will separate $[T_0, 1]$ into two intervals, $[T_0, 3T_*]$ and $[T_*, 1]$, where $T_* := N^{-(\kappa^{-1} - \delta)/d}$, and provide different upper bounds. The reason for this choice of division point $T_*$ is sketched as follows and is detailed in C.2. In the approximation of the vector field, we use the $B$-spline approximation of densities as in Oko et al. (2023). To show a fast convergence rate, the first interval is more subtle because $p_t(x)$ is rougher. In approximating the density on the smoother boundary region, we divide the region into small cubes, each of which uses $N^\delta$ bases for $B$-spline approximation. To make the total number of $B$-spline bases comparable with $N$, the width $a_0$ of the region should be $a_0 = N^{(1-\kappa\delta)/d}$. On the other hand, in Theorem 7, we need a concentration of an integral around the boundary region for a better approximation by the higher smoothness, and this limits the variance of the Gaussian kernel so that $\sigma_t = t^\kappa \le a_0$. This derives $t \le N^{-(\kappa^{-1}-\delta)/d}$. As a result, the division point is small enough as $T_* := N^{-(\kappa^{-1}-\delta)/d}$.

The approximation bound for $t \in [T_0, 3T_*]$ with $T_* := N^{-\frac{\kappa^{-1}-\delta}{d}}$ is given in the following Theorem (see Appendix C.4 for the proof).

**Theorem 7.** *Under assumptions (A1)-(A5), there is a neural network $\phi_1 \in \mathcal{M}(L, W, S, B)$ and a constant $C_6$, which is independent of $t$, such that, for sufficiently large $N$,*

$$\int \|\phi_1(\boldsymbol{x}, t) - \boldsymbol{v}_t(\boldsymbol{x})\|^2 p_t(\boldsymbol{x}) d\boldsymbol{x} \le C_6 \{(\sigma_t')^2 \log N + (m_t')^2\} N^{-\frac{2s}{d}}, \tag{20}$$

*for any $t \in [T_0, 3T_*]$, where $L = O(\log^4 N)$, $\|W\|_\infty = O(N \log^6 N)$, $S = O(N \log^8 N)$, $B = \exp(O(\log N \log \log N))$. Additionally, we can take $\phi_1$ to satisfy*

$$\|\phi_1(\boldsymbol{x}, t)\| \le \tilde{C}_6 \{(\sigma_t')\sqrt{\log n} + |m_t'|\},$$

*where $\tilde{C}_6$ is a constant independent of $t$.*

### 4.2.3 Approximation error for large $t$

We derive a bound of the approximation error on any interval $[2t_*, 1]$, where $t_* \ge T_* = N^{-\frac{\kappa^{-1}-\delta}{d}}$. This is used to discuss the optimal convergence rate in Section 4.3.

**Theorem 8.** *Fix $t_* \in [T_*, 1]$ and take an arbitrary $\eta > 0$. Under Assumptions (A1)-(A5), there is a neural network $\phi_2 \in \mathcal{M}(L, W, S, B)$ and $C_7 > 0$, which does not depend on $t$, such that the bound*

$$\int \|\phi_2(\boldsymbol{x}, t) - \boldsymbol{v}_t(\boldsymbol{x})\|^2 p_t(\boldsymbol{x}) d\boldsymbol{x} \le C_7 \{(\sigma_t')^2 \log N + (m_t')^2\} N^{-\eta} \tag{21}$$

*holds for all $t \in [2t_*, 1]$, and the NN model satisfies $L = O(\log^4 N)$, $\|W\|_\infty = O(N)$, $S = O(t_*^{-d\kappa} N^{\delta\kappa})$, and $B = \exp(O(\log N \log \log N)$. Moreover, $\phi_2$ can be taken so that*

$$\|\phi_2(\cdot, t)\|_\infty \le \tilde{C}_7 \{(\sigma_t') \log N + |m_t'|\}$$

*with constant $\tilde{C}_7 > 0$ independent of $t$.*

See Appendix C.5 for the proof. The approximation error $N^{-\eta}$ is arbitrarily small and is not dominant, while $S$ may be dominant in the complexity term.

### 4.3 Convergence rate under Wasserstein distance

We can consider a generalization bound based on Theorems 3, 4, 7, and 8 deriving the bounds for $[T_0, 2T_*]$ and $[2T_*, 1]$. However, if we apply Theorem 8 for $[2t_*, 1]$ with $t_* = T_* = N^{-(\kappa^{-1}-\delta)/d}$, the dominant factor of the log covering number in (17) is the sparsity $S = O(t_*^{-d\kappa} N^{\delta\kappa}) = O(N)$. From Theorems 3 and 4, the complexity part gives $O((N/n)^{1/2})$ term in the $W_2$ generalization error. If we plug $N = n^{(2s+d)/d}$, which is optimal for the MSE generalization, we have $O(n^{-s/(2s+d)})$ for the upper bound of $W_2$ generalization, which is slower than the lower bound in Proposition 2. To

achieve a better convergence rate, we will make use of the factor $\sqrt{t}$ in front of (13) by dividing the interval $[T_0, 1]$ into small pieces and using a NN for each small interval, as in Oko et al. (2023) for diffusion models.

Notice that, when time $t$ is far from 0, the convolution of $p_t(\boldsymbol{x}|\boldsymbol{y})$ with larger $\sigma_t$ results in a smoother target vector field $\boldsymbol{v}_t(\boldsymbol{x})$, which is easier to approximate with a low-complexity model. On the other hand, when $t$ approaches 0, with the fixed number of $B$-spline bases $N$, the approximation error bound $\{(\sigma_t')^2\sqrt{\log N} + (m_t')^2\}N^{-2s/d}$ can increase for large $\sigma_t'$ or $m_t'$ (e.g. $\sigma_t \sim t^\kappa$ with $\kappa < 1$). We therefore need a more complex model (that is, larger $N$) than the one needed for larger $t$. Thus, it will be more efficient if the number of $B$-spline bases $N$, which controls the approximation error and complexity, is chosen adaptively to the time region $t$.

Specifically, we make a partition $T_0 = t_0 < t_1 < t_2 < \cdots < t_K = 1$ such that $t_j = 2t_{j-1}$ for $1 \le j \le K - 1$ with $2t_{K-1} \ge 1$, and build a neural network for each $[t_{j-1}, t_j]$ $(j = 1, \ldots, K)$. Note that we train each network for interval $[t_{j-1}, t_j]$ with $n$ training data $(\boldsymbol{x}_i)_{i=1}^n$. We assume that $t_{j_*}$ with $T_* \le t_{j_*} \le 3T_*$ serves as the boundary to apply two different error bounds. The total number of intervals $K$ is $O(\log N) = O(\log n)$, since $2^K T_0 \ge 1$ with $T_0 = N^{-R_0}$ can be achieved by $K \ge R_0 \log_2 N$. The constant $R_0$ is fixed as $R_0 \ge \frac{s+1}{\min(\kappa, \bar{\kappa})}$ so that $W_2(P_{T_0}, P_0)$ is negligible (see the proof sketch of Theorem 9). In this setting, we have the following main result.

**Theorem 9** (Main result). *Assume (A1)-(A5) and $d \ge 2$. If the above time-partition is applied and a neural network is trained for each time division, for arbitrarily small $\delta > 0$ and $1 \le r \le 2$, we have*

$$\mathbb{E}\left[W_r(\widehat{P}_{T_0}, P_{true]})\right] = O\left(n^{-\frac{s+(2\kappa)^{-1}-\delta}{2s+d}}\right) \qquad (n \to \infty). \tag{22}$$

*Proof Sketch.* Let $J_j := [t_{j-1}, t_j]$ $(j = 1, \ldots, K)$. We use a smaller neural network model for larger $j$. Specifically, the number of $B$-spline bases for $J_j$ is $N_j' := t_{j-1}^{-d\kappa} N^{\delta\kappa}$ for $j > j_*$, while $N_j' = N$ for $j \le j_*$, where $j_*$ is defined as above. Note that $N_j' \to \infty$ as $N \to \infty$ due to $\delta > 0$, and that $N_j' \le N^{1-\delta\kappa} N^{\delta\kappa} = N$ for $j \ge j_*$ due to $t_j \ge N^{-\frac{\kappa^{-1}-\delta}{d}}$, which means a lower complexity. See also Figure E.1 in Appendix.

Next, we consider the bound of $W_2$-distance based on the partition. For each of $j = 1, \ldots, K$, we introduce a vector field $\tilde{\boldsymbol{v}}_t^{(j)}$ such that it coincides with the target $\boldsymbol{v}_t$ for $t \in [t_j, 1]$ and with the learned $\widehat{\boldsymbol{v}}_t$ for $t \in [T_0, t_j]$. Let $Q^{(j)}$ be the pushforward from $P_1 = N_d(0, I_d)$ to $t = T_0$ by the flow of the vector field $\tilde{\boldsymbol{v}}_t^{(j)}$. Then, $Q^{(0)} = P_{T_0}$, the pushforward by the flow of the target $\boldsymbol{v}_t$ from $t = 1$ to $T_0$, and also $Q^{(K)} = \widehat{p}_{T_0}$. Note also that $\tilde{\boldsymbol{v}}^{(j)}$ and $\tilde{\boldsymbol{v}}^{(j-1)}$ differ only in $J_j$ by $\boldsymbol{v}_t(\boldsymbol{x}) - \widehat{\boldsymbol{v}}_t(\boldsymbol{x})$. Therefore, from Theorem 3 and the Lipschitz assumption on $\mathcal{H}_n$, we have

$$W_2(P_0, \hat{P}_{T_0}) \le W_2(P_0, P_{T_0}) + \sum_{j=1}^K W_2(Q^{(j-1)}, Q^{(j)})$$

$$\le \theta_n + C \sum_{j=1}^K \sqrt{t_j} \left\{ \int_{t_{j-1}}^{t_j} e^{2\int_t^{t_j}(\bar{C}/u)du} \int \|\hat{\phi}(\boldsymbol{x}, t) - \boldsymbol{v}(\boldsymbol{x}, t)\|^2 p_t(\boldsymbol{x}) d\boldsymbol{x} dt \right\}^{1/2},$$

where $\theta_n^2 = db_0^2 n^{-\frac{2R_0\kappa}{2s+d}} + \int \|y\|^2 dP_0(\boldsymbol{y}) \tilde{b}_0^2 n^{-\frac{2R_0\tilde{\kappa}}{2s+d}}$, which is derived from Lemma 11 and (A3). We take a constant $R_0 \ge (s+1)/\min(\kappa, \bar{\kappa})$ so that $\theta_n$ is of $O(n^{-\frac{s+1}{2s+d}})$ and thus $\theta_n$ is negligible. It is easy to see that the factor $e^{\int_t^{t_j}(\bar{C}/u)du}$ is bounded by a constant because of $t_j = 2t_{j-1}$ by definition.

For simplicity, let $t_* := t_{j_*}$. From Theorems 7, 8, and 4, the generalization bound of $\hat{P}_{T_0}$ is given by

$$\mathbb{E}\left[W_2(P_0, \hat{P}_{T_0})\right]$$

$$\le \theta_n + \sum_{j=1}^{j_*} \sqrt{t_*} \left\{ C_6 \int_{t_{j-1}}^{t_j} \{(\sigma_t')^2 \log N + (m_t')^2\} N^{-2s/d} dt + \frac{N}{n} O(\text{poly}(\log n)) \right\}^{1/2}$$

$$+ \sum_{j=k_*}^K \sqrt{t_j} \left\{ C_7 \int_{t_{j-1}}^{t_j} \{(\sigma_t')^2 \log N + (m_t')^2\} N^{-\eta} dt + \frac{t_j^{-d\kappa} N^{\delta\kappa}}{n} O(\text{poly}(\log n)) \right\}^{1/2}$$

$$\le \theta_n + C''' \sqrt{t_*} t_*^{\kappa-1/2} N^{-s/d} O(\text{poly}(\log n)) + C''' \sqrt{\frac{N}{n}} O(\text{poly}(\log n))$$

$$+ C''' \sum_{j=k_*}^K \left\{ \sqrt{t_j} N^{-\eta/2} O(\text{poly}(\log n)) + \sqrt{t_j} \frac{t_j^{-d\kappa/2} N^{\delta\kappa/2}}{\sqrt{n}} O(\text{poly}(\log n)) \right\}$$

$$\leq \theta_n + C'''t_*^\kappa n^{-s/(2s+d)}O(\text{poly}(\log n)) + C'''\sqrt{t_*}n^{-s/(2s+d)}O(\text{poly}(\log n))$$
$$+C'''\sum_{j=k_*}^{K}\left\{\sqrt{t_j}N^{-\eta/2}O(\text{poly}(\log n)) + t_j^{\frac{1-d\kappa}{2}}n^{\frac{d\delta\kappa}{2(2s+d)}-\frac{1}{2}}O(\text{poly}(\log n))\right\}.$$

In the third inequality, we use $\int_{t_{j-1}}^{t_j}\{(\sigma_t')^2 + (m_t')^2\}dt = O(\text{poly}(\log n))$ for $\kappa = 1/2$, and the fact that it is bounded by a constant for $\kappa > 1/2$. Since $\eta$ is arbitrarily large and $\kappa \geq 1/2$, neglecting the factors of $\text{poly}(\log n)$, the candidates of the dominant terms in the above expression are $t_*^{1/2}n^{-\frac{s}{2s+d}}$ in the third term and $t_*^{\frac{1-d\kappa}{2}}n^{\frac{d\delta\kappa}{2(2s+d)}-\frac{1}{2}}$ in the last summation. By balancing these two terms, the upper bound can be minimized by setting

$$t_* = C_*n^{-\frac{\kappa^{-1}-\delta}{2s+d}}, \tag{23}$$

for some contact $C_*$, and the dominant term of the upper-bound is given by

$$\tilde{O}\left(n^{-\frac{s+(2\kappa)^{-1}-\delta/2}{2s+d}}\right). \tag{24}$$

This proves the claim by replacing $\delta/2$ with $\delta$ and absorbing the $\text{poly}(\log)$ factor in $\delta$. $\qquad\square$

From Proposition 2 and Theorem 9, if $\kappa = 1/2$, the FM method achieves an almost optimal rate up to the $\text{poly}(\log n)$ factor and arbitrary small $\delta > 0$. On the other hand, for $\kappa > 1/2$, the obtained upper bound is not optimal. This suggests that the choice of $\sigma_t \sim \sqrt{t}$ around $t \to 0^+$ is theoretically reasonable. This is also a popular choice for the diffusion model.

## 4.4 DISCUSSION

In the derivation of the almost minimax optimal convergence rate, the use of neural networks for divided time intervals is a limitation of the current theoretical analysis. As discussed before Theorem 9, without this partition, the current analysis gives only $\tilde{O}(n^{-\frac{s}{2s+d}})$, which is not optimal for $W_2$. It is obviously an important question of how to avoid such a time division. In Oko et al. (2023), the optimal convergence rate for the diffusion model has been proved without time division for the total variation, which is $\tilde{O}(n^{-\frac{s}{2s+d}})$. The bound is based on Girsanov's theorem, which gives an upper bound of the KL divergence of SDE by the $L_2$ losses of the drift estimation. To the best of our knowledge, no bounds are known for ODE with respect to the KL or total variation for the difference of vector fields. This is an important future direction for understanding the ability of FM theoretically.

## 5 CONCLUSION

This paper has rigorously analyzed the convergence rate of flow matching, demonstrating for the first time that FM can achieve the almost minimax optimal convergence rate under the 2-Wasserstein distance. This result positions FM as a competitive alternative to diffusion models in terms of asymptotic convergence rates, which concurs with empirical results in various applications. Our findings further reveal that the convergence rate is significantly influenced by the variance decay rate in the Gaussian conditional kernel, where $\sigma_t \sim \sqrt{t}$ is shown to yield the optimal rate. Although there are several popular proposals for the mean and variance functions, theoretical justification or comparison has not been explored intensively. The current result on the upper bound (Theorem 9) provides theoretical insight on the influence of the choice of these functions.

Although this study offers substantial theoretical contributions, these insights are still grounded in specific modeling assumptions that limit broader applicability. In addition to the time-partition discussed in Sec. 4.4, this paper focuses primarily on assumptions utilizing Gaussian conditional kernels. However, other FM implementations might employ different path constructions, as suggested by recent proposals Kerrigan et al. (2023); Isobe et al. (2024). The theoretical implications of these alternative approaches remain an essential area for future research.

### ACKNOWLEDGMENTS

This work has been supported in part by JST CREST JPMJCR2015 and JSPS Grant-in-Aid for Transformative Research Areas (A) 22H05106.

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

# Appendix

This section summarizes some basic mathematical facts, which can be easily derived and used in the proof of our main results, and known facts developed in Oko et al. (2023).

## A    BASIC MATHEMATICAL RESULTS

### A.1    DEFINITION OF BESOV SPACE

Besov space is an extension of the Sobolev space, allowing for non-integer orders of smoothness, and is effective in measuring both the local regularity and the global behavior of functions. It is formally defined as follows.

Let $\Omega$ be a domain in $\mathbb{R}^d$. For a function $f \in L_{p'}(\Omega)$ for some $p' \in (0, \infty]$, the *r-th modulus of smoothness* of $f$ is defined by

$$w_{r,p'}(f,t) = \sup_{\|\boldsymbol{h}\|_2 \leq t} \|\Delta_{\boldsymbol{h}}^r(f)\|_{p'},$$

where

$$\Delta_{\boldsymbol{h}}^r(f)(\boldsymbol{x}) = \begin{cases} \sum_{j=0}^r \binom{r}{j}(-1)^{r-j} f(\boldsymbol{x}+j\boldsymbol{h}) & \text{if } \boldsymbol{x}+j\boldsymbol{h} \in \Omega \text{ for all } j, \\ 0 & \text{otherwise.} \end{cases}$$

For $0 < p', q' \leq \infty, s > 0, r := |s| + 1$, let the seminorm $|\cdot|_{B_{p',q'}^s}$ be defined by

$$|f|_{B_{p',q'}^s} := \begin{cases} \left(\int_0^\infty (t^{-s} w_{r,p'}(f,t))^{q'} \frac{dt}{t}\right)^{\frac{1}{q'}} & (q' < \infty), \\ \sup_{t>0} t^{-s} w_{r,p'}(f,t) & (q' = \infty). \end{cases}$$

The norm of the *Besov space* $B_{p',q'}^s(\Omega)$ is defined by

$$\|f\|_{B_{p',q'}^s} := \|f\|_{p'} + |f|_{B_{p',q'}^s},$$

and

$$B_{p',q'}^s(\Omega) := \{f \in L_{p'}(\Omega) \mid \|f\|_{B_{p',q'}^s} < \infty\}.$$

The parameter $s$ serves as the order of smoothness. If $p' = q'$ and $s$ is an integer, $B_{p',q'}^s$ coincides with the Sobolev space. For details of Besov spaces, see Triebel (1992), for example.

### A.2    LIPSCHITZ CONDITION

This section shows a lemma that provides a Lipschitz constant of $\boldsymbol{x} \mapsto \boldsymbol{v}_t(\boldsymbol{x})$ under the assumptions (A3) and (A5).

**Lemma 10.** *Let $\boldsymbol{v}_t(\boldsymbol{x})$ be a vector field defined by (3) and (8), i.e.,*

$$\boldsymbol{v}_t(\boldsymbol{x}) = \int \left\{\sigma_t' \frac{\boldsymbol{x} - m_t\boldsymbol{y}}{\sigma_t} + m_t'\boldsymbol{y}\right\} p_t(\boldsymbol{y}|\boldsymbol{x}) d\boldsymbol{y}$$

*where*

$$p_t(\boldsymbol{y}|\boldsymbol{x}) = \frac{p_t(\boldsymbol{x}|\boldsymbol{y})p_0(\boldsymbol{y})}{\int p_t(\boldsymbol{x}|\tilde{\boldsymbol{y}})p_0(\tilde{\boldsymbol{y}})d\tilde{\boldsymbol{y}}}, \quad p_t(\boldsymbol{x}|\boldsymbol{y}) = \frac{1}{(\sqrt{2\pi}\sigma_t)^d} e^{-\frac{\|\boldsymbol{x}-m_t\boldsymbol{y}\|^2}{2\sigma_t^2}}.$$

*Then, under (A3) and (A5), $\boldsymbol{v}_t(\boldsymbol{x})$ is Lipschitz continuous with Lipschitz constant $\tilde{C}_L/t$ for any sufficiently small $t$, where $\tilde{C}_L$ is independent of $t$.*

*Proof.* By the definition of $\boldsymbol{v}_t$ and the form of $\sigma_t$ and $m_t$ in (A3), we can compute explicitly

$$\frac{\partial \boldsymbol{v}_t(\boldsymbol{x})}{\partial \boldsymbol{x}} = \frac{\kappa}{t} I_d + \int \left\{\frac{\kappa}{t}(\boldsymbol{x} - m_t\boldsymbol{y}) + \frac{\bar{\kappa}}{t}\boldsymbol{y}\right\} \frac{\partial p_t(\boldsymbol{y}|\boldsymbol{x})}{\partial \boldsymbol{x}} d\boldsymbol{y}.$$

As $\frac{\partial}{\partial \boldsymbol{x}} \int p_t(\boldsymbol{y}|\boldsymbol{x}) d\boldsymbol{y} = \frac{\partial}{\partial \boldsymbol{x}} 1 = 0$, we further obtain

$$\frac{\partial \boldsymbol{v}_t(\boldsymbol{x})}{\partial \boldsymbol{x}} = \frac{\kappa}{t} I_d + \frac{\bar{\kappa} - m_t\kappa}{t} \frac{\partial}{\partial \boldsymbol{x}} \int \boldsymbol{y} p_t(\boldsymbol{y}|\boldsymbol{x}) d\boldsymbol{y}.$$

The claim is obvious under (A5). $\qquad\square$

## A.3 WASSERSTEIN DISTANCE FOR CONVOLUTION

The following lemma is used in the proof sketch of Theorem 9, where $W_2(P_t, P_{T_0})$ is bounded.

**Lemma 11.** *Let $P$ be a probability distribution on $\mathbb{R}^d$ with $V := \int \|\boldsymbol{y}\|^2 dP(\boldsymbol{y}) < \infty$ and $P_{m,\sigma}$ be given by the density $\int \frac{1}{(\sqrt{2\pi}\sigma)^d} \exp(-\frac{\|\boldsymbol{x}-m\boldsymbol{y}\|^2}{2\sigma^2}) dP(\boldsymbol{y})$. Then,*

$$W_2(P, P_{m,\sigma}) \le \sqrt{(1-m)^2 V + d\sigma^2}.$$

*Proof.* The proof is elementary, but we include it for completeness. Let $Y$ and $Z$ be independent random variables with probability $P$ and $N_d(0, I_d)$, respectively. Let $X := mY + \sigma Z$, then the distribution of $X$ is $P_{m,\sigma}$. Considering a coupling $(X, Y)$,

$$
\begin{aligned}
W_2(P, P_{m,\sigma})^2 &\le E\|X - Y\|^2 \\
&= E\|(m-1)Y + \sigma Z\|^2 \\
&= (1-m)^2 V + d\sigma^2,
\end{aligned}
$$

which completes the proof. □

## A.4 APPROXIMATION OF A FUNCTION IN BESOV SPACE

In this subsection, we present several approximation results developed in Suzuki (2019); Oko et al. (2023). Although these results are already known, we include them here for ease of reference.

Let $\mathcal{N}(x)$ be the function defined by $\mathcal{N}(x) = 1$ for $x \in [0, 1]$ and 0 otherwise. The *cardinal B-spline* of order $\ell \in \mathbb{N}$ is defined by

$$\mathcal{N}_\ell(x) := \mathcal{N} * \mathcal{N} * \cdots * \mathcal{N}(x),$$

which is the convolution $(\ell + 1)$ times of $\mathcal{N}$. Here, the convolution $f * g$ is defined by

$$(f * g)(x) = \int f(x - y)g(y)dy.$$

For a multi-index $k \in \mathbb{N}^d$ and $j \in \mathbb{Z}^d$, the *tensor product B-spline basis* in $\mathbb{R}^d$ of order $\ell$ is defined by

$$M_{k,j}^d(\boldsymbol{x}) := \prod_{i=1}^d \mathcal{N}_\ell(2^{k_i} x_i - j_i).$$

The following theorem says that a function $f$ in the Besov space is approximated by a superposition of $M_{k,j}^d(\boldsymbol{x})$ of the form

$$f_N(\boldsymbol{x}) = \sum_{(k,j)} \alpha_{k,j} M_{k,j}^d(\boldsymbol{x}).$$

In the sequel, we fix the order $\ell$ of the B-spline. $(a)_+$ denotes $\max\{0, a\}$.

**Theorem 12** (Oko et al. (2023); Suzuki (2019)). *Let $C > 0$ and $0 < p', q', r \le \infty$. Under $s > d(1/p' - 1/r)_+$ and $0 < s < \min\{\ell, \ell - 1 + 1/p'\}$, where $\ell \in \mathbb{N}$ is the order of the cardinal B-spline bases, for any $f \in B_{p',q'}^s([-C, C]^d)$, there exists $f_N$ that satisfies*

$$\|f - f_N\|_{L_r([-C,C]^d)} \lesssim C^s N^{-s/d} \|f\|_{B_{p',q'}^s([-C,C]^d)}$$

*for $N \gg 1$ and has the following form:*

$$f_N(\boldsymbol{x}) = \sum_{k=0}^K \sum_{j \in J(k)} \alpha_{k,j} M_{k,j}^d(\boldsymbol{x}) + \sum_{k=K+1}^{K^*} \sum_{i=1}^{n_k} \alpha_{k,j_i} M_{k,j_i}^d(\boldsymbol{x})$$

*with*

$$\sum_{k=0}^K |J(k)| + \sum_{k=K+1}^{K^*} n_k = N,$$

*where $J(k) = \{-C2^k - \ell, -C2^k - \ell + 1, \ldots, C2^k - 1, C2^k\}$, $(j_i)_{i=1}^{n_k} \subset J(k)$, $K = O(d^{-1} \log(N/C^d))$, $K^* = (O(1) + \log(N/C^d))\nu^{-1} + K$, $n_k = O((N/C^d)2^{-\nu(k-K)})$ $(k = K+1, \ldots, K^*)$ for $\nu = (s - \omega)/(2\omega)$ with $\omega = d(1/p' - 1/r)_+$. Moreover, we can take $\alpha_{k,j}$ so that $|\alpha_{k,j}| \le N^{(\nu^{-1}+d^{-1})(d/p'-s)_+}$.*

Based on the above theorem, the following result shows the accuracy of approximating the true density $p_0$ by the $B$-spline functions with support restriction.

**Theorem 13** (Oko et al. (2023)). *Under Assumptions (A1)-(A5), there exists $f_N$ of the form*

$$f_N(\boldsymbol{x}) = \sum_{i=1}^{N} \alpha_i \mathbf{1}\left[\|\boldsymbol{x}\|_\infty \leq 1\right] M_{k_i,j_i}^d(\boldsymbol{x}) + \sum_{i=N+1}^{3N} \alpha_i \mathbf{1}\left[\|\boldsymbol{x}\|_\infty \leq 1 - N^{-\frac{\kappa^{-1}-\delta}{d}}\right] M_{k_i,j_i}^d(\boldsymbol{x}), \quad (25)$$

*that satisfies*

$$\|p_0 - f_N\|_{L^2(I^d)} \leq C_a N^{-s/d}, \quad (26)$$

$$\|p_0 - f_N\|_{L^2(I^d \setminus I_N^d)} \leq C_a N^{-\check{s}/d}, \quad (27)$$

*for some $C_a > 0$ and $f_N(\boldsymbol{x}) = 0$ for any $x$ with $\|\boldsymbol{x}\|_\infty \geq 1$. Here, $-2^{(k_i)_m} - \ell \leq (j_i)_m \leq 2^{(k_i)_m}$ ($i = 1, 2, \ldots, N$; $m = 1, 2, \ldots, d$), $|k_i| \leq K^* = (O(1) + \log N)\nu^{-1} + O(d^{-1}\log N)$ for $\nu = (2s - \omega)/(2\omega)$ with $\omega = d(1/p - 1/2)_+$. The notations $(k_i)_m$ and $(j_i)_m$ are the m-th component of the multi-indices $k_i$ and $j_i$, respectively. Moreover, we can take $|\alpha_i| \leq N(\nu^{-1} + d^{-1})(d/p - s)_+$.*

*Proof.* See Oko et al. (2023, Lemma B.4). $\qquad\square$

The following result shows the accuracy of approximating the "smoothed" $B$-spline basis function by a neural network. This is essential to consider the approximation of $p_t(\boldsymbol{x})$ and $\boldsymbol{v}_t(\boldsymbol{x})$ by a neural network taken for $p_0(\boldsymbol{x})$.

**Theorem 14.** *Let $C > 0$, $k \in \mathbb{Z}_+$, $j \in \mathbb{Z}^d$, $\ell \in \mathbb{Z}_+$ with $-C2^k - \ell \leq j_i \leq C2^k$ ($i = 1, 2, \ldots, d$), and $C_{b,1} = 1$ or $1 - N^{-(1-\delta)}$. For any $\varepsilon$ ($0 < \varepsilon < 1/2$), there exists a neural network $\phi_3^{k,j}(\boldsymbol{x}, t)$ and $\phi_4^{k,j}(\boldsymbol{x}, t)$ in $\mathcal{M}(L, W, S, B)$ with*

$$L = O(\log^4 \varepsilon^{-1} + \log^2 C + k^2),$$

$$\|W\|_\infty = O(\log^6 \varepsilon^{-1} + \log^3 C + k^3),$$

$$S = O(\log^8 \varepsilon^{-1} + \log^4 C + k^4),$$

$$B = \exp(O(\log \varepsilon^{-1} \log\log \varepsilon^{-1} + \log C + k)) \quad (28)$$

*such that*

$$\left| \phi_3^{k,j}(\boldsymbol{x}, t) - \int_{\mathbb{R}^d} \mathbf{1}\left[\|y\|_\infty \leq C_{b,1}\right] M_{k,j}^d(\boldsymbol{y}) \frac{1}{(\sqrt{2\pi}\sigma_t)^d} e^{-\frac{\|\boldsymbol{x} - m_t \boldsymbol{y}\|^2}{2\sigma_t^2}} d\boldsymbol{y} \right| \leq \varepsilon \quad (29)$$

*and*

$$\left| \phi_4^{k,j}(\boldsymbol{x}, t) - \int_{\mathbb{R}^d} \frac{\boldsymbol{x} - m_t \boldsymbol{y}}{\sigma_t} \mathbf{1}\left[\|\boldsymbol{y}\|_\infty \leq C_{b,1}\right] M_{k,j}^d(\boldsymbol{y}) \frac{1}{(\sqrt{2\pi}\sigma_t)^d} e^{-\frac{\|\boldsymbol{x} - m_t \boldsymbol{y}\|^2}{2\sigma_t^2}} d\boldsymbol{y} \right| \leq \varepsilon \quad (30)$$

*hold for all $\boldsymbol{x} \in [-C, C]^d$. Furthermore, we can choose the networks so that $\|\phi_3^{k,j}\|_\infty$ and $\|\phi_4^{k,j}\|_\infty$ are of class $O(1)$.*

*Proof.* See Oko et al. (2023, Lemma B.3) $\qquad\square$

### A.5 APPROXIMATION OF GAUSSIAN INTEGRALS

The following lemma is an elementary fact about Gaussian integrals used in the proof of Theorem 7 in Section C.4. We include it here for completeness.

**Lemma 15.** *Let $\boldsymbol{x} \in \mathbb{R}^d$, $0 < \varepsilon < 1/2$, and $\alpha \in \{0, 1\}$. For any function $F(\boldsymbol{y})$ supported on $I^d$, there is $C_b > 0$ that depends only on $d$ such that*

$$\left| \int_{I^d} \frac{1}{(\sqrt{2\pi}\sigma_t)^d} e^{-\frac{\|\boldsymbol{x} - m_t \boldsymbol{y}\|^2}{2\sigma_t^2}} F(\boldsymbol{y}) d\boldsymbol{y} - \int_{A_{\boldsymbol{x}}} \frac{1}{(\sqrt{2\pi}\sigma_t)^d} e^{-\frac{\|\boldsymbol{x} - m_t \boldsymbol{y}\|^2}{2\sigma_t^2}} F(\boldsymbol{y}) d\boldsymbol{y} \right| \leq \varepsilon, \quad (31)$$

*where*

$$A_{\boldsymbol{x}} := \left\{ \boldsymbol{y} \in I^d \mid \left\| \boldsymbol{y} - \frac{\boldsymbol{x}}{m_t} \right\|_\infty \leq C_b \frac{\sigma_t \sqrt{\log N}}{m_t} \right\}.$$

*Proof.* See Oko et al. (2023, Lemma F.9). $\qquad\square$

## B   PROOF OF THEOREM 3

Although the proof of Theorem 3 is basically the same as the proof of Benton et al. (2023b, Theorem 1), a slight difference appears since the current bound shows for arbitrary time $t$. We include the proof here for completeness.

Let $\boldsymbol{v}(\boldsymbol{x}, t)$ and $\widehat{\boldsymbol{v}}(\boldsymbol{x}; t)$ be smooth vector fields and $\boldsymbol{\varphi}_{s,t}$ and $\widehat{\boldsymbol{\varphi}}_{s,t}$ be respective flows;

$$\frac{d}{dt}\boldsymbol{\varphi}_{s,t}(\boldsymbol{x}) = \boldsymbol{v}(\boldsymbol{\varphi}_{s,t}(\boldsymbol{x}), t), \quad \boldsymbol{\varphi}_{s,s}(\boldsymbol{x}) = \boldsymbol{x},$$

$$\frac{d}{dt}\widehat{\boldsymbol{\varphi}}_{s,t}(\boldsymbol{x}) = \widehat{\boldsymbol{v}}(\widehat{\boldsymbol{\varphi}}_{s,t}(\boldsymbol{x}), t), \quad \widehat{\boldsymbol{\varphi}}_{s,s}(\boldsymbol{x}) = \boldsymbol{x}$$

**Lemma 16** (Alekseev-Gröbner). *Under the above notations, for any $T \geq 0$,*

$$\widehat{\boldsymbol{\varphi}}_{0,T}(\boldsymbol{x}_0) - \boldsymbol{\varphi}_{0,T}(\boldsymbol{x}_0) = \int_0^T \left(\nabla_x \widehat{\boldsymbol{\varphi}}_{s,T}(\boldsymbol{x})\big|_{x=\boldsymbol{\varphi}_{0,s}(\boldsymbol{x}_0)}\right)\left(\widehat{\boldsymbol{v}}(\boldsymbol{\varphi}_{0,s}(\boldsymbol{x}_0), s) - \boldsymbol{v}(\boldsymbol{\varphi}_{0,s}(\boldsymbol{x}_0), s)\right)ds. \quad (32)$$

Note that on the right-hand side, the vector fields $\widehat{\boldsymbol{v}}$ and $\boldsymbol{v}$ are evaluated at the same point $\boldsymbol{\varphi}_{0,s}(\boldsymbol{x}_0)$.

*Proof of Theorem 3.* Let $\widehat{P}_t := (\widehat{\boldsymbol{\varphi}}_{0,t})_\# P_0$ and $P_t := (\boldsymbol{\varphi}_{0,t})_\# P_0$ be pushforwards. By the definition of the 2-Wasserstein metric,

$$W_2(\widehat{P}_t, P_t) \leq \left(\int \|\widehat{\boldsymbol{\varphi}}_{0,t}(\boldsymbol{x}_0) - \boldsymbol{\varphi}_{0,t}(\boldsymbol{x}_0)\|^2 dP_0(\boldsymbol{x}_0)\right)^{1/2}. \quad (33)$$

From Lemma 16,

$$\|\widehat{\boldsymbol{\varphi}}_{0,t}(\boldsymbol{x}_0) - \boldsymbol{\varphi}_{0,t}(\boldsymbol{x}_0)\| \leq \int_0^t \left\|\nabla_x \widehat{\boldsymbol{\varphi}}_{s,t}(\boldsymbol{x})\big|_{x=\boldsymbol{\varphi}_{0,s}(\boldsymbol{x}_0)}\right\|_{op} \|\widehat{\boldsymbol{v}}(\boldsymbol{\varphi}_{0,s}(\boldsymbol{x}_0), s) - \boldsymbol{v}(\boldsymbol{\varphi}_{0,s}(\boldsymbol{x}_0), s)\| \, ds.$$

As a general relation of the largest singular value, we have

$$\frac{\partial}{\partial t}\left\|\nabla_x \widehat{\boldsymbol{\varphi}}_{s,t}(\boldsymbol{x})\right\|_{op} \leq \left\|\frac{\partial}{\partial t}\nabla_x \widehat{\boldsymbol{\varphi}}_{s,t}(\boldsymbol{x})\right\|_{op}.$$

Then, it follows from $\frac{\partial}{\partial t}\nabla_x \widehat{\boldsymbol{\varphi}}_{s,t}(\boldsymbol{x}) = \nabla_y \widehat{\boldsymbol{v}}(y, t)|_{y=\boldsymbol{\varphi}_{s,t}(\boldsymbol{x})}\nabla_x \widehat{\boldsymbol{\varphi}}_{s,t}(\boldsymbol{x})$ that the inequality

$$\frac{\partial}{\partial t}\left\|\nabla_x \widehat{\boldsymbol{\varphi}}_{s,t}(\boldsymbol{x})\big|_{x=\boldsymbol{\varphi}_{0,s}(\boldsymbol{x}_0)}\right\|_{op} \leq L_t \left\|\nabla_x \boldsymbol{\varphi}_{s,t}(\boldsymbol{x})|_{x=\boldsymbol{\varphi}_{0,s}(\boldsymbol{x}_0)}\right\|$$

holds by the $L_t$-Lipschitzness of $\widehat{\boldsymbol{v}}(\cdot, t)$. Accordingly, noting that $\|\nabla_x \widehat{\boldsymbol{\varphi}}_{s,s}(\boldsymbol{x})\|_{op} = \|\nabla_x \boldsymbol{x}\|_{op} = 1$, the standard ODE argument leads to

$$\left\|\nabla_x \widehat{\boldsymbol{\varphi}}_{s,t}(\boldsymbol{x})\big|_{x=\boldsymbol{\varphi}_{0,s}(\boldsymbol{x}_0)}\right\|_{op} \leq e^{\int_s^t L_u du},$$

and therefore

$$\|\widehat{\boldsymbol{\varphi}}_{0,t}(\boldsymbol{x}_0) - \boldsymbol{\varphi}_{0,t}(\boldsymbol{x}_0)\| \leq \int_0^t e^{\int_s^t L_u du}\|\widehat{\boldsymbol{v}}(\boldsymbol{\varphi}_{0,s}(\boldsymbol{x}_0), s) - \boldsymbol{v}(\boldsymbol{\varphi}_{0,s}(\boldsymbol{x}_0), s)\|ds.$$

From Cauchy-Schwarz inequality,

$$\|\widehat{\boldsymbol{\varphi}}_{0,t}(\boldsymbol{x}_0) - \boldsymbol{\varphi}_{0,t}(\boldsymbol{x}_0)\|^2$$
$$\leq \int_0^t 1^2 ds \int_0^t e^{2\int_s^t L_u du}\|\widehat{\boldsymbol{v}}(\boldsymbol{\varphi}_{0,s}(\boldsymbol{x}_0), s) - \boldsymbol{v}(\boldsymbol{\varphi}_{0,s}(\boldsymbol{x}_0), s)\|^2 ds$$
$$= t \int_0^t e^{2\int_s^t L_u du}\|\widehat{\boldsymbol{v}}(\boldsymbol{\varphi}_{0,s}(\boldsymbol{x}_0), s) - \boldsymbol{v}(\boldsymbol{\varphi}_{0,s}(\boldsymbol{x}_0), s)\|^2 ds.$$

Since the distribution of $\boldsymbol{\varphi}_{0,s}(\boldsymbol{x}_0)$ with $x_0 \sim P_0$ is given by $P_s$, combining the above bound with (33) completes the proof. $\square$

## C  PROOF OF MAIN THEOREMS

### C.1  PROOF OF LEMMA 5: SUP OF LOSS FUNCTIONS

*Proof.*

$$\ell_{\phi}(\boldsymbol{x}) = \int_{T_{\ell}}^{T_u} \int \|\phi(\boldsymbol{x}_t, t) - \boldsymbol{v}_t(\boldsymbol{x}_t|\boldsymbol{x})\|^2 p_t(\boldsymbol{x}_t|\boldsymbol{x}) d\boldsymbol{x}_t dt$$

$$\leq 2 \int_{T_{\ell}}^{T_u} \int \|\phi(\boldsymbol{x}_t, t)\|^2 p_t(\boldsymbol{x}_t|\boldsymbol{x}) d\boldsymbol{x}_t dt + 2 \int_{T_{\ell}}^{T_u} \int \|\boldsymbol{v}_t(\boldsymbol{x}_t|\boldsymbol{x})\|^2 p_t(\boldsymbol{x}_t|\boldsymbol{x}) d\boldsymbol{x}_t dt$$

From the definition of $\mathcal{H}_n$ and Assumptions (A3-A4), the first term is bounded by

$$4C^2 \int_{T_0}^{1} \left( (\sigma_t')^2 \log n + (m_t')^2 \right) dt \leq 8\tilde{C} \left( (\log n)^{b+1} + (\log n)^b \right),$$

where $\tilde{C} > 0$ is a constant, and $b = 0$ for $\kappa > 1/2$ and $b = b_0$ for $\kappa = 1/2$. From the definition

$$\boldsymbol{v}_t(\boldsymbol{x}_t|\boldsymbol{x}) = \sigma_t' \frac{\boldsymbol{x}_t - m_t \boldsymbol{x}}{\sigma_t} + m_t' \boldsymbol{x}$$

and the fact $\|\boldsymbol{x}\| \leq 1$, the second term is upper bounded by

$$4 \int_{T_0}^{1} \int \left\{ (\sigma_t')^2 \frac{\|\boldsymbol{x}_t - m_t \boldsymbol{x}\|^2}{\sigma_t^2} + (m_t')^2 \|\boldsymbol{x}\|^2 \right\} p_t(\boldsymbol{x}_t|\boldsymbol{x}) d\boldsymbol{x}_t dt$$

$$\leq 4 \int_{T_0}^{1} \left\{ d(\sigma_t')^2 + (m_t')^2 \right\} dt \quad \leq \quad 4d\tilde{D}(\log n)^b$$

for some constants $\tilde{D} > 0$. Note that the first inequality is given by $p_t(\boldsymbol{x}_t|\boldsymbol{x}) = N_d(m_t \boldsymbol{x}, \sigma_t^2 I_d)$. This completes the proof. $\square$

### C.2  DIVISION POINT

The appropriate division point of the time interval $[T_0, 1]$ arises from the width $a_0$ of the smoother region around the boundary of $I^d$, which is assumed in (A1). Suppose that the smoothness of $p_0(x)$ is $\check{s}$, higher than $s$, in the region $I^d \backslash [-1 + a_0, 1 - a_0]^d$. We should make the smoother region as small as possible to ensure that $p_0$ is essentially in $B_{p,1}^s(I^d)$. We set $a_0 = N^{-\gamma}$ and consider the partition of $I^d \backslash [-1 + a_0, 1 - a_0]^d$ by $N^{d\gamma} - (N^{\gamma} - 2)^d$ cubes of size $a_0 = N^{-\gamma}$ (see Figure C.1). Suppose that we use $N^{\delta'}$ bases ($N \gg 1$) of $B$-spline for each small cube with arbitrarily small $\delta' > 0$. The condition $\delta' > 0$ guarantees that the approximation can be arbitrarily accurate in each small cube. Since $p_0$ restricted to each cube has a smooth degree $\check{s}$, Theorem 12 tells that it can be approximated by a $B$-spline function with the accuracy

$$a_0^{-\check{s}} N^{-\check{s}\delta'/d}.$$

The total number of $B$-spline bases is then

$$(N^{d\gamma} - (N^{\gamma} - 2)^d)N^{\delta'} \sim N^{(d-1)\gamma + \delta'},$$

To make the number of bases equal to or less than $N$, which is the number used for the $B_{p',q'}^s$-region of $p_0$, we set $\gamma = (1 - \delta\kappa)/d$, i.e., $a_0 = N^{-(1-\delta\kappa)/d}$ (we set $\delta' = \delta\kappa$ for notational simplicity).

As seen in the proof of Theorem 7, to obtain the desired bound, the deviation $\sigma_t$ should satisfy $\sigma_t \leq a_0$ to bound the integral around the boundary. When $t$ is small so that $\sigma_t \sim t^{\kappa}$, $\sigma_t \leq a_0$ means $t \lesssim T_* := N^{-(\kappa^{-1}-\delta)/d}$, which gives a constant on $T_*$. Consequently, we divide the time interval into $[T_0, T_*]$ and $[T_*, 1]$, and show the different bounds for the approximation error.

### C.3  BASIC BOUNDS OF $p_t(x)$ AND $v_t(x)$

Recall that

$$p_t(\boldsymbol{x}) = \int_{[-1,1]^d} \frac{1}{(2\pi\sigma_t)^d} e^{-\frac{\|\boldsymbol{x} - m_t \boldsymbol{y}\|^2}{2\sigma_t^2}} p_0(\boldsymbol{y}) d\boldsymbol{y}.$$

$a_0$

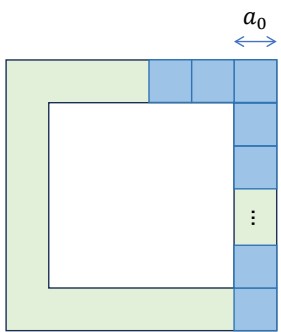

Figure C.1: Division of the cube into smoother small regions and the general region.

**Lemma 17.** *There exists $C_1 = C_1(d, C_0) > 0$ such that*

$$C_1^{-1} \exp\Big(-\frac{(\|\boldsymbol{x}\|_\infty - m_t)_+^2}{\sigma_t^2}\Big) \le p_t(\boldsymbol{x}) \le C_1 \exp\Big(-\frac{(\|\boldsymbol{x}\|_\infty - m_t)_+^2}{2\sigma_t^2}\Big) \tag{34}$$

*for any $\boldsymbol{x} \in \mathbb{R}^d$ and $t \in [T_0, 1]$.*

*Proof.* The proof is elementary and the same as Oko et al. (2023, Lemma A.2). We omit it. □

**Lemma 18.** *(i) Let $k \in \mathbb{N}$ be arbitrary. There is $C_2 = C_2(d, k, C_0) > 0$ such that*

$$\Big\|\partial_{x_{i_1}} \cdots \partial_{x_{i_k}} p_t(\boldsymbol{x})\Big\| \le \frac{C_2}{\sigma_t^k} \quad (\forall t \in [T_0, 1]).$$

*(ii) There is $C_3 = C_3(d, C_0) > 0$ such that*

$$\|\boldsymbol{v}_t(\boldsymbol{x})\| \le C_3\Big\{\sigma_t'\Big(\frac{(\|\boldsymbol{x}\|_\infty - m_t)_+}{\sigma_t} \vee 1\Big) + |m_t'|\Big\}$$

*for any $\boldsymbol{x} \in \mathbb{R}^d$ and $t \in [T_0, 1]$.*

*Proof.* (i) is standard and the same as (Lemma A.3 Oko et al., 2023). We omit it.

For (ii), let $g(\boldsymbol{x}; m_t\boldsymbol{y}, \sigma_t^2) = p_t(\boldsymbol{x}|\boldsymbol{y}) = \frac{1}{(2\pi\sigma_t)^{d/2}} \exp\big\{-\frac{\|x - m_t\boldsymbol{y}\|^2}{2\sigma_t^2}\big\}$. Note that

$$v_t(\boldsymbol{x}) = \frac{\int v_t(\boldsymbol{x}|\boldsymbol{y}) g(\boldsymbol{x}; m_t\boldsymbol{y}, \sigma_t^2) p_0(\boldsymbol{y}) d\boldsymbol{y}}{p_t(\boldsymbol{x})}.$$

The norm of the numerator is upper bounded by

$$\Big\|\int \Big\{\sigma_t' \frac{\boldsymbol{x} - m_t\boldsymbol{y}}{\sigma_t} + m_t' y\Big\} g(\boldsymbol{x}; m_t\boldsymbol{y}, \sigma_t^2) p_0(\boldsymbol{y}) d\boldsymbol{y}\Big\|$$

$$\le \sigma_t' \Big\|\int \frac{\boldsymbol{x} - m_t\boldsymbol{y}}{\sigma_t} g(\boldsymbol{x}; m_t\boldsymbol{y}, \sigma_t^2) p_0(\boldsymbol{y}) d\boldsymbol{y}\Big\| + |m_t'| \int \|\boldsymbol{y}\| g(\boldsymbol{x}; m_t\boldsymbol{y}, \sigma_t^2) p_0(\boldsymbol{y}) d\boldsymbol{y}.$$

Since $\text{Supp}(p_0) = [-1, 1]^d$ by assumption (A1), the second term in the last line is upper bounded by

$$|m_t'| \int g(\boldsymbol{x}; m_t\boldsymbol{y}, \sigma_t^2) p_0(\boldsymbol{y}) d\boldsymbol{y} = |m_t'| p_t(\boldsymbol{x}). \tag{35}$$

To bound the first term, we use the restriction of the integral region as in Lemma F.9, Oko et al. (2023). Namely, letting $\varepsilon := C_1^{-1} \exp(-\frac{(\|\boldsymbol{x}\|_\infty - m_t)_+^2}{2\sigma_t^2})$, the lower bound of $p_t(\boldsymbol{x})$, the integral is approximated as for any $j = 1, \ldots, d$,

$$\Big|\int_{\mathbb{R}^d} \Big(\frac{x_j - m_t y_j}{\sigma_t}\Big)^\alpha g(\boldsymbol{x}; m_t\boldsymbol{y}, \sigma_t^2) p_0(\boldsymbol{y}) d\boldsymbol{y} - \int_{A_x} \Big(\frac{x_j - m_t y_j}{\sigma_t}\Big)^\alpha g(\boldsymbol{x}; m_t\boldsymbol{y}, \sigma_t^2) p_0(\boldsymbol{y}) d\boldsymbol{y}\Big| \le \varepsilon,$$

where $A_x := \prod_{i=1}^d \left[ \frac{x_i}{m_t} - \frac{C\sigma_c}{m_t}\sqrt{\log(1/\varepsilon)}, \frac{x_i}{m_t} + \frac{C\sigma_t}{m_t}\sqrt{\log(1/\varepsilon)} \right]$, $C$ is a positive constant depending only on $d$, and $\alpha \in \{0, 1\}$. Then, the first term is upper-bounded by

$$\sigma_t' \left\| \int_{A_x} \frac{x - m_t y}{\sigma_t} g(x; m_t y, \sigma_t^2) p_0(y) dy + \varepsilon \mathbf{1} \right\|.$$

Noting that $y \in A_x$ is equivalent to $|x_i - m_t y_i|/\sigma_t \leq C\sqrt{\log(1/\varepsilon)}$, the above quantity is further upper-bounded by

$$\sqrt{d} C \sigma_t' \sqrt{\log(1/\varepsilon)} \int_{A_x} g(x; m_t y, \sigma_t^2) p_0(y) dy + \sqrt{d}\varepsilon \leq C'\left( \sigma_t' \sqrt{\log(1/\varepsilon)} p_t(x) + \varepsilon \right).$$

Noting that $\varepsilon = C_1^{-1} \exp\left(-\frac{(\|x\|_\infty - m_t)_+^2}{\sigma_t^2}\right)$ and $p_t(x) \geq \varepsilon$, we obtain

$$\|v_t(x)\| \leq \frac{C'(\sigma_t' \sqrt{\log(1/\varepsilon)} p_t(x) + \varepsilon) + |m_t'| p_t(x)}{p_t(x)}$$

$$\leq C'' \left\{ \sigma_t' \left( \frac{(\|x\|_\infty - m_t)_+}{\sigma_t} \right) + \sigma_t' + |m_t'| \right\}$$

$$\leq C''' \left\{ \sigma_t' \left( \frac{(\|x\|_\infty - m_t)_+}{\sigma_t} \vee 1 \right) + |m_t'| \right\}.$$

This completes the proof. $\square$

The following lemma shows an upper bound of $v_t(x)$ when $x$ is in a bounded region of $\sigma_t\sqrt{1/\varepsilon}$, and presents bounds of relevant integrals.

**Lemma 19.** *Let $\varepsilon > 0$ be sufficiently small.*

*(i) For any $C_4 > 0$, we have*

$$\|v_t(x)\| \leq C_4\left(\sigma_t'\sqrt{\log(1/\varepsilon)} + |m_t'|\right) \tag{36}$$

*for any $x$ with $\|x\|_\infty \leq m_t + C_4\sigma_t\sqrt{\log(1/\varepsilon)}$ and $t \in [T_0, 1]$.*

*(ii) For any $C_5 > 0$, there is $\tilde{C} > 0$ such that*

$$\int_{\|x\|_\infty \geq m_t + C_5\sigma_t\sqrt{\log(1/\varepsilon)}} p_t(x) \|v_t(x)\|^2 dx \leq \tilde{C}\left\{ (\sigma_t')^2 \log^{\frac{d}{2}}\left(\varepsilon^{-1}\right) + (m_t')^2 \log^{\frac{d-2}{2}}\left(\varepsilon^{-1}\right) \right\} \varepsilon^{\frac{C_5^2}{2}} \tag{37}$$

*and*

$$\left| \int_{\|x\|_\infty \geq m_t + C_5\sigma_t\sqrt{\log(1/\varepsilon)}} p_t(x) dx \right| \leq \tilde{C} \log^{\frac{d-2}{2}}\left(\varepsilon^{-1}\right) \varepsilon^{\frac{C_5^2}{2}} \tag{38}$$

*hold for any $\varepsilon > 0$ and $t \in [T_0, 1]$.*

*Proof.* (i) is obvious from Lemma 18, since $(\|x\|_\infty - m_t)_+/\sigma_t \leq C_4\sqrt{\log(1/\varepsilon)}$ under the assumption of $x$.

We show (ii). It follows from Lemmas 17 and 18 that

$$p_t(x)\|v_t(x)\|^2 \leq 2C_1 C_3 \exp\left(-\frac{(\|x\|_\infty - m_t y)_+^2}{2\sigma_t^2}\right)\left\{ (\sigma_t')^2 \frac{(\|x\|_\infty - m_t)_+^2}{\sigma_t^2} + (m_t')^2 \right\}.$$

Let $r := (\|x\|_\infty - m_t)_+/\sigma_t$ and $B_i := \{x = (x_1, \ldots, x_d) \in \mathbb{R}^d \mid |x_i| = \max_{1 \leq j \leq d} |x_j|\}$. The space is divided into $d$ regions $\cup_{i=1}^d B_i$ with measure-zero intersections. In $B_1$, the variables $x_2, \ldots, x_d$ satisfy $|x_j| \leq m_t + C_4\sigma_t\sqrt{\log(1/\varepsilon)}$, and integral (37) is upper bounded by

$$2C_1 C_3 d \int_{C_4\sqrt{\log(1/\varepsilon)}} e^{-\frac{r^2}{2}} \left((\sigma_t')^2 r^2 + (m_t')^2\right)(\sigma_t r + m_t)^{d-1} dr$$

$$\leq C' \int_{C_4\sqrt{\log(1/\varepsilon)}} e^{-\frac{r^2}{2}} \left((\sigma_t')^2 r^{d+1} + (m_t')^2 r^{d-1}\right) dr,$$

where we use $(\sigma_t r + m_t)^{d-1} \le (r+1)^{d-1} \le 2^{d-1} r^{d-1}$.

For $\ell \in \mathbb{N} \cup \{0\}$, define

$$\psi_\ell(z) := \int_z^\infty r^\ell e^{-\frac{r^2}{2}} \, dr.$$

It is easy to see $\psi_1(z) = e^{-\frac{z^2}{2}}$ and $\psi_\ell(z) = x^{\ell-1} e^{-\frac{z^2}{2}} + (\ell-1)\psi_{\ell-1}(z)$ by partial integral. Using these formulas, we can see that for $z \ge 1$

$$\psi_\ell(z) \le B_\ell z^{\ell-1} e^{-\frac{z^2}{2}},$$

where $B_\ell$ is a positive constant that depends only on $\ell$.

Thus, we obtain an upper bound

$$\tilde{C}\big\{(\sigma_t')^2 \log^{\frac{d}{2}}\big(\frac{1}{\varepsilon}\big) + (m_t')^2 \log^{\frac{d-2}{2}}\big(\frac{1}{\varepsilon}\big)\big\}\varepsilon^{\frac{C_5^2}{2}},$$

which proves (37). The assertion (38) is similar. $\qquad\qquad\square$

## C.4 Bounds of the approximation error for small $t$

This subsection shows the proof of Theorem 7.

**(I) Restriction of the integral.**

We first show that the left-hand side of (20) can be approximated by the integral over the bounded region

$$D_{t,N} := \{\boldsymbol{x} \in \mathbb{R}^d \mid \|\boldsymbol{x}\|_\infty \le m_t + C_4\sigma_t\sqrt{\log N}\}.$$

To see this, observe that from Lemma 18 (ii), for $\boldsymbol{x} \in D_{t,N}$ we have

$$\|\boldsymbol{v}_t(\boldsymbol{x})\|^2 \le 2C_3^2\left\{(\sigma_t')^2 \log N + |m_t'|^2\right\}. \tag{39}$$

From the bound of $\boldsymbol{v}_t(\boldsymbol{x})$, we can restrict the neural network $\boldsymbol{\phi}(\boldsymbol{x}, t)$ so that it satisfies the same upper bound. Therefore, (39) is applied to $\|\boldsymbol{\phi}(\boldsymbol{x}, t)\|^2$ also. Combining this fact with Lemma 19 (ii), we have

$$\int_{D_{t,N}^C} \|\boldsymbol{\phi}(\boldsymbol{x}, t) - \boldsymbol{v}_t(\boldsymbol{x})\|^2 p_t(\boldsymbol{x}) d\boldsymbol{x}$$

$$\le 4C_3^2\left\{(\sigma_t')^2 \log N + |m_t'|^2\right\} \int_{D_{t,N}^C} \frac{1}{(\sqrt{2\pi}\sigma)^d} e^{-\frac{\|\boldsymbol{x} - m_t\boldsymbol{y}\|^2}{2\sigma^2}} d\boldsymbol{x}$$

$$\le 4C_3^2\tilde{C}\left\{(\sigma_t')^2 \log N + |m_t'|^2\right\} N^{-C_4^2/2} \log^{\frac{d-2}{2}} N.$$

If $C_4$ is taken large enough to satisfy $C_4^2/2 > \frac{2s}{d}$, it follows that

$$\int \|\boldsymbol{\phi}(\boldsymbol{x}, t) - \boldsymbol{v}_t(\boldsymbol{x})\|^2 p_t(\boldsymbol{x}) d\boldsymbol{x}$$

$$\le \int_{D_{t,N}} \|\boldsymbol{\phi}(\boldsymbol{x}, t) - \boldsymbol{v}_t(\boldsymbol{x})\|^2 p_t(\boldsymbol{x}) d\boldsymbol{x} + C'\left\{(\sigma_t')^2 \log N + |m_t'|^2\right\} N^{-\frac{2s}{d}} \tag{40}$$

for some constant $C' > 0$. Thus, we can consider the first term on the right-hand side.

Let $\omega > 0$ be an arbitrary positive number. The integral over $D_{t,N}$ in (40) can be restricted to the region $\{\boldsymbol{x} \mid p_t(\boldsymbol{x}) \ge N^{-\frac{2s+\omega}{d}}\}$. This can be easily seen by

$$\int_{D_{t,N}} \mathbf{1}[p_t(\boldsymbol{x}) \le N^{-\frac{2s+\omega}{d}}]\|\boldsymbol{v}_t(\boldsymbol{x}) - \boldsymbol{\phi}(\boldsymbol{x}, t)\|^2 p_t(\boldsymbol{x}) d\boldsymbol{x}$$

$$\le 4C_3^2 \int_{D_{t,N}} \left\{(\sigma_t')^2 \log N + |m_t'|^2\right\} N^{-\frac{2s+\omega}{d}} d\boldsymbol{x}$$

$$\le 4C_3^2 N^{-\frac{2s+\omega}{d}} \left\{(\sigma_t')^2 \log N + |m_t'|^2\right\} 2^d(m_t + C_4\sigma_t\sqrt{\log N})^d$$

$$\le C''\left\{(\sigma_t')^2 \log N + |m_t'|^2\right\} N^{-\frac{2s+\omega}{d}} \log^{\frac{d}{2}} N,$$

where $C''$ depends only on $d$, $C_3$, and $C_4$. This bound is of smaller order than the second term on the right-hand side of (40), and thus negligible. In summary, we have

$$\int \|\phi(\boldsymbol{x}, t) - \boldsymbol{v}_t(\boldsymbol{x})\|^2 p_t(\boldsymbol{x}) d\boldsymbol{x}$$

$$\leq \int_{D_{t,N}} \mathbf{1}[p_t(\boldsymbol{x}) \geq N^{-\frac{2s+\omega}{d}}] \|\phi(\boldsymbol{x}, t) - \boldsymbol{v}_t(\boldsymbol{x})\|^2 p_t(\boldsymbol{x}) d\boldsymbol{x} + C' \left\{ (\sigma_t')^2 \log N + |m_t'|^2 \right\} N^{-\frac{2s}{d}}$$

$$(41)$$

for sufficiently large $N$.

**(II) Decomposition of integral.** Here, we give a bound of the integral $\int \|\phi(\boldsymbol{x}, t) - \boldsymbol{v}_t(\boldsymbol{x})\|^2 p_t(\boldsymbol{x}) d\boldsymbol{x}$ over the region $D_{t,N} \cap \{\boldsymbol{x} \mid p_t(\boldsymbol{x}) \geq N^{-\frac{2s+\omega}{d}}\}$. The norm $\|\phi(\boldsymbol{x}, t) - \boldsymbol{v}_t(\boldsymbol{x})\|$ is bounded in detail. First, recall that

$$\boldsymbol{v}_t(\boldsymbol{x}) = \frac{\int \boldsymbol{v}_t(\boldsymbol{x}|\boldsymbol{y}) p_t(\boldsymbol{x}|\boldsymbol{y}) p_0(\boldsymbol{y}) d\boldsymbol{y}}{p_t(\boldsymbol{x})}, \quad p_t(\boldsymbol{x}) = \int p_t(\boldsymbol{x}|\boldsymbol{y}) p_0(\boldsymbol{y}) d\boldsymbol{y}. \tag{42}$$

Based on Theorem 13, we can find $f_N$ in (25) such that

$$\|p_0 - f_N\|_{L^2(I^d)} \leq C_a N^{-s/d}, \qquad \|p_0 - f_N\|_{L^2(I^d \setminus I_N^d)} \leq C_a N^{-\check{s}/d} \tag{43}$$

As an approximate of $p_t(\boldsymbol{x})$, define a function $\tilde{f}_1(\boldsymbol{x}, t)$ by

$$\tilde{f}_1(\boldsymbol{x}, t) := \int \frac{1}{(\sqrt{2\pi}\sigma_t)^d} e^{-\frac{\|\boldsymbol{x} - m_t \boldsymbol{y}\|^2}{2\sigma_t^2}} f_N(\boldsymbol{y}) d\boldsymbol{y}. \tag{44}$$

Since we consider the region where $p_t(\boldsymbol{x}) \geq N^{-(2s+\omega)/d}$, we further define

$$f_1(\boldsymbol{x}, t) := \tilde{f}_1(\boldsymbol{x}, t) \vee N^{-\frac{2s+\omega}{d}}.$$

In a similar manner, the numerator of (42) can be approximated by

$$\sigma_t' \boldsymbol{f}_2(\boldsymbol{x}, t) + m_t' \boldsymbol{f}_3(\boldsymbol{x}, t),$$

where $\mathbb{R}^d$-valued functions $\boldsymbol{f}_2$ and $\boldsymbol{f}_3$ are defined by

$$\boldsymbol{f}_2(\boldsymbol{x}, t) := \int \frac{\boldsymbol{x} - m_t \boldsymbol{y}}{\sigma_t} \frac{1}{(\sqrt{2\pi}\sigma_t)^d} e^{-\frac{\|\boldsymbol{x} - m_t \boldsymbol{y}\|^2}{2\sigma_t^2}} f_N(\boldsymbol{y}) d\boldsymbol{y}.$$

$$\boldsymbol{f}_3(\boldsymbol{x}, t) := \int \boldsymbol{y} \frac{1}{(\sqrt{2\pi}\sigma_t)^d} e^{-\frac{\|\boldsymbol{x} - m_t \boldsymbol{y}\|^2}{2\sigma_t^2}} f_N(\boldsymbol{y}) d\boldsymbol{y}. \tag{45}$$

We have an approximate of $\boldsymbol{v}_t(\boldsymbol{x})$ by

$$\boldsymbol{f}_4(\boldsymbol{x}, t) := \frac{\sigma_t' \boldsymbol{f}_2(\boldsymbol{x}, t) + m_t' \boldsymbol{f}_3(\boldsymbol{x}, t)}{f_1(\boldsymbol{x}, t)} \mathbf{1}\left[\left|\frac{\boldsymbol{f}_2}{f_1}\right| \leq C_5 \sqrt{\log N}\right] \mathbf{1}\left[\left|\frac{\boldsymbol{f}_3}{f_1}\right| \leq C_5\right].$$

We want to evaluate

$$\int_{D_{t,N}} \mathbf{1}[p_t(\boldsymbol{x}) \geq N^{-\frac{2s+\omega}{d}}] \|\phi(\boldsymbol{x}, t) - \boldsymbol{v}_t(\boldsymbol{x})\|^2 p_t(\boldsymbol{x}) d\boldsymbol{x}$$

$$\leq \int_{D_{t,N}} \mathbf{1}[p_t(\boldsymbol{x}) \geq N^{-\frac{2s+\omega}{d}}] \|\phi(\boldsymbol{x}, t) - \boldsymbol{f}_4(\boldsymbol{x}, t)\|^2 p_t(\boldsymbol{x}) d\boldsymbol{x}$$

$$+ \int_{D_{t,N}} \mathbf{1}[p_t(\boldsymbol{x}) \geq N^{-\frac{2s+\omega}{d}}] \|\boldsymbol{f}_4(\boldsymbol{x}, t) - \boldsymbol{v}_t(\boldsymbol{x})\|^2 p_t(\boldsymbol{x}) d\boldsymbol{x}$$

$$=: I_A + I_B. \tag{46}$$

**(III) Bound of $I_A$ (neural network approximation of B-spline)**

We will approximate $f_1$, $\boldsymbol{f}_2$, and $\boldsymbol{f}_3$ by neural networks. For $k \in \mathbb{Z}_+$ and $j \in \mathbb{Z}^d$, let $E^{(a)}_{k,j,u}$ ($a = 1, 2, 3$, $u = 0, 1$) denote the functions defined by

$$E^{(1)}_{k,j,u} := \int_{\mathbb{R}^d} \mathbf{1}\left[\|\boldsymbol{y}\|_\infty \leq C_{b,1}\right] M^d_{k,j}(\boldsymbol{y}) \frac{1}{(\sqrt{2\pi}\sigma_t)^d} e^{-\frac{\|\boldsymbol{x} - m_t \boldsymbol{y}\|^2}{2\sigma_t^2}} d\boldsymbol{y}, \tag{47}$$

$$E^{(2)}_{k,j,u} := \int_{\mathbb{R}^d} \frac{\boldsymbol{x} - m_t \boldsymbol{y}}{\sigma_t} \mathbf{1}\left[\|\boldsymbol{y}\|_\infty \leq C_{b,1}\right] M^d_{k,j}(\boldsymbol{y}) \frac{1}{(\sqrt{2\pi}\sigma_t)^d} e^{-\frac{\|\boldsymbol{x} - m_t \boldsymbol{y}\|^2}{2\sigma_t^2}} d\boldsymbol{y}, \tag{48}$$

and

$$E^{(3)}_{k,j,u} := \int_{\mathbb{R}^d} \boldsymbol{y} \mathbf{1}\left[\|\boldsymbol{y}\|_\infty \leq C_{b,1}\right] M^d_{k,j}(\boldsymbol{y}) \frac{1}{(\sqrt{2\pi}\sigma_t)^d} e^{-\frac{\|\boldsymbol{x} - m_t \boldsymbol{y}\|^2}{2\sigma_t^2}} d\boldsymbol{y}, \tag{49}$$

where $C_{b,1} = 1$ for $u = 0$ and $C_{b,1} = 1 - N^{-\frac{\kappa^{-1} - \delta}{d}}$ for $u = 1$. Then, from Theorem 12, $f_N$ is written as a linear combination of $\mathbf{1}[\|\boldsymbol{y}\|_\infty \leq C_{b,1}]M^d_{k,j}$ with coefficients $\alpha_{k,j}$. From Theorem 13, for any $\varepsilon > 0$, there are neural networks $\phi_5$, $\phi_6$ and $\phi_7$ such that

$$|f_1(\boldsymbol{x}, t) - \phi_5(\boldsymbol{x}, t)| \leq D_5 N \max_i |\alpha_i| \varepsilon$$

$$\|\boldsymbol{f}_2(\boldsymbol{x}, t) - \boldsymbol{\phi}_6(\boldsymbol{x}, t)\| \leq D_6 N \max_i |\alpha_i| \varepsilon,$$

$$\|\boldsymbol{f}_3(\boldsymbol{x}, t) - \boldsymbol{\phi}_7(\boldsymbol{x}, t)\| \leq D_7 N \max_i |\alpha_i| \varepsilon.$$

Since $\max_i |\alpha_i| \leq N^{-(\nu^{-1} + d^{-1})(d/p - s)_+}$, by taking $\varepsilon$ sufficently small, for any $\eta > 0$ we have

$$|f_1(\boldsymbol{x}, t) - \phi_5(\boldsymbol{x}, t)| \leq D_5 N^{-\eta}$$

$$\|\boldsymbol{f}_2(\boldsymbol{x}, t) - \boldsymbol{\phi}_6(\boldsymbol{x}, t)\| \leq D_6 N^{-\eta}$$

$$\|\boldsymbol{f}_3(\boldsymbol{x}, t) - \boldsymbol{\phi}_7(\boldsymbol{x}, t)\| \leq D_7 N^{-\eta}.$$

The operations to obtain the approximation of $\boldsymbol{v}_t(\boldsymbol{x})$ based on $\phi_5$, $\boldsymbol{\phi}_6$, and $\boldsymbol{\phi}_7$ are given by the following procedures:

$$\zeta_1 := \texttt{clip}(\phi_5; N^{-(2s+\omega)/d}, N^{K_0+1}),$$

$$\zeta_2 := \texttt{recip}(\zeta_1),$$

$$\boldsymbol{\zeta}_3 := \texttt{mult}(\zeta_2, \boldsymbol{\phi}_6),$$

$$\boldsymbol{\zeta}_4 := \texttt{clip}(\boldsymbol{\zeta}_3; -C_5 \sqrt{\log N}, C_5 \sqrt{\log N}),$$

$$\boldsymbol{\zeta}_5 := \texttt{mult}(\zeta_2, \boldsymbol{\phi}_7),$$

$$\boldsymbol{\zeta}_6 := \texttt{clip}(\boldsymbol{\zeta}_5; -C_5, C_5),$$

$$\boldsymbol{\zeta}_7 := \texttt{mult}(\boldsymbol{\zeta}_4, \hat{\sigma}'),$$

$$\boldsymbol{\zeta}_8 := \texttt{mult}(\boldsymbol{\zeta}_6, \hat{m}'),$$

$$\boldsymbol{\phi}_8 := \boldsymbol{\zeta}_7 + \boldsymbol{\zeta}_8.$$

As shown in Section D, the neural networks $\texttt{clip}$, $\texttt{recip}$, and $\texttt{mult}$ achieve the approximation error $N^{-\eta}$ with arbitrarily large $\eta$, while the complexity of the networks increases only at most polynomials of $N$ for $B$ and $\|W\|_\infty$, and at most $\text{poly}(\log N)$ factor for $L$ and $S$. In the upper bound of the generalization error, the network parameters $B$ and $\|W\|_\infty$ and the inverse error $\varepsilon^{-1} = N^\eta$ appear only in the $\log()$ part to the log covering number, and $S$ and $B$ appear as a linear factor. We also need to use the approximations $\hat{\sigma}'_t$ and $\hat{m}'_t$ of $\sigma'_t$ and $m'_t$, respectively, by neural networks in construction. However, this can be done in a similar manner to (Section B1, Oko et al., 2023) with all network parameters $O(\log^r \varepsilon^{-1})$ for approximation accuracy $\varepsilon$, and thus they have only $O(\text{poly}(\log N))$ contributions. We omit the details in this paper. Consequently, the increase of the neural networks to obtain $\boldsymbol{\phi}_8$ from $\phi_5$, $\boldsymbol{\phi}_6$, and $\boldsymbol{\phi}_7$ contributes the log covering number only by the $\text{poly}(\log N)$ factor.

As a result, we obtain

$$I_A = O(N^{-\eta}\text{poly}(\log N)) \tag{50}$$

for arbitrary $\eta > 0$, while the required neural network increases the complexity term only by $O(\text{poly}(\log N))$ factor.

**(IV) Bound of $I_B$ (B-spline approximation of the true vector field)**

We evaluate here

$$I_B = \int_{D_{t,N}} \mathbf{1}[p_t(\boldsymbol{x}) \geq N^{-\frac{2s+\omega}{d}}] \|\boldsymbol{f}_4(\boldsymbol{x}, t) - \boldsymbol{v}_t(\boldsymbol{x})\|^2 p_t(\boldsymbol{x}) d\boldsymbol{x}. \tag{51}$$

Let $\boldsymbol{h}_2(\boldsymbol{x}, t)$ and $\boldsymbol{h}_3(\boldsymbol{x}, t)$ be functions defined by

$$\boldsymbol{h}_2(\boldsymbol{x}, t) := \int_{\mathbb{R}^d} \frac{\boldsymbol{x} - m_t \boldsymbol{y}}{\sigma_t} \frac{1}{(\sqrt{2\pi}\sigma_t)^d} e^{-\frac{\|\boldsymbol{x} - m_t \boldsymbol{y}\|^2}{2\sigma_t^2}} p_0(\boldsymbol{y}) d\boldsymbol{y},$$

$$\boldsymbol{h}_3(\boldsymbol{x}, t) := \int_{\mathbb{R}^d} \boldsymbol{y} \frac{1}{(\sqrt{2\pi}\sigma_t)^d} e^{-\frac{\|\boldsymbol{x} - m_t \boldsymbol{y}\|^2}{2\sigma_t^2}} p_0(\boldsymbol{y}) d\boldsymbol{y}. \tag{52}$$

Then,

$$\|\boldsymbol{f}_4(\boldsymbol{x}, t) - \boldsymbol{v}_t(\boldsymbol{x})\|$$
$$= \mathbf{1}\left[\left\|\frac{\boldsymbol{f}_2}{f_1}\right\| \leq C_5 \sqrt{\log N}\right] \mathbf{1}\left[\left\|\frac{\boldsymbol{f}_3}{f_1}\right\| \leq C_5\right] \left\|\frac{\sigma_t' \boldsymbol{f}_2(\boldsymbol{x}, t) + m_t' \boldsymbol{f}_3(\boldsymbol{x}, t)}{f_1(\boldsymbol{x})} - \frac{\sigma_t' \boldsymbol{h}_2(\boldsymbol{x}, t) + m_t' \boldsymbol{h}_3(\boldsymbol{x}, t)}{p_t(\boldsymbol{x})}\right\|. \tag{53}$$

We evaluate the integral (51) by dividing $D_{t,N}$ into the two domains $\{\boldsymbol{x} \mid \|\boldsymbol{x}\|_\infty \leq m_t\}$ and $\{\boldsymbol{x} \mid m_t \leq \|\boldsymbol{x}\|_\infty \leq m_t + C_4 \sigma_t \sqrt{\log N}\}$.

Due to condition $p_t(\boldsymbol{x}) \geq N^{-(2s+\omega)/d}$, it suffices to take the network $f_1(\boldsymbol{x}, t)$ so that it satisfies $f_1(\boldsymbol{x}, t) \geq N^{-(2s+\omega)/d}$ by clipping the function if necessary. We therefore assume in the sequel that $f_1(\boldsymbol{x}, t) \geq N^{-(2s+\omega)/d}$ holds.

**(IV-a) case: $\|\boldsymbol{x}\|_\infty \leq m_t$.**

In this case, Lemma 17 shows that $C_1^{-1} \leq p_t(x) \leq C_1$ for some $C_1 > 0$, which depends only on $d$ and $p_0$. Using this fact and the conditions $\left\|\frac{\boldsymbol{f}_2}{f_1}\right\| \leq C_5 \sqrt{\log N}$ and $\left\|\frac{\boldsymbol{f}_3}{f_1}\right\| \leq C_5$, we have

$$\left\|\frac{\sigma_t' \boldsymbol{f}_2(\boldsymbol{x}, t) + m_t' \boldsymbol{f}_3(\boldsymbol{x}, t)}{f_1(\boldsymbol{x})} - \frac{\sigma_t' \boldsymbol{h}_2(\boldsymbol{x}, t) + m_t' \boldsymbol{h}_3(\boldsymbol{x}, t)}{p_t(\boldsymbol{x})}\right\|$$

$$\leq |\sigma_t'| \left\|\frac{\boldsymbol{f}_2(\boldsymbol{x}, t)}{f_1(\boldsymbol{x})} - \frac{\boldsymbol{h}_2(\boldsymbol{x}, t)}{p_t(\boldsymbol{x})}\right\| + |m_t'| \left\|\frac{\boldsymbol{f}_3(\boldsymbol{x}, t)}{f_1(\boldsymbol{x})} - \frac{\boldsymbol{h}_3(\boldsymbol{x}, t)}{p_t(\boldsymbol{x})}\right\|$$

$$\leq |\sigma_t'| \left\{\left\|\frac{\boldsymbol{f}_2(x)}{f_1(\boldsymbol{x})} - \frac{\boldsymbol{f}_2(\boldsymbol{x}, t)}{p_t(\boldsymbol{x})}\right\| + \left\|\frac{\boldsymbol{f}_2(\boldsymbol{x}, t)}{p_t(\boldsymbol{x})} - \frac{\boldsymbol{h}_2(\boldsymbol{x}, t)}{p_t(\boldsymbol{x})}\right\|\right\}$$

$$\quad + |m_t'| \left\{\left\|\frac{\boldsymbol{f}_3(x)}{f_1(\boldsymbol{x})} - \frac{\boldsymbol{f}_3(\boldsymbol{x}, t)}{p_t(\boldsymbol{x})}\right\| + \left\|\frac{\boldsymbol{f}_3(\boldsymbol{x}, t)}{p_t(\boldsymbol{x})} - \frac{\boldsymbol{h}_3(\boldsymbol{x}, t)}{p_t(\boldsymbol{x})}\right\|\right\}$$

$$\leq C_1 |\sigma_t'| \left\{C_5 \sqrt{\log N} |p_t(\boldsymbol{x}) - f_1(\boldsymbol{x}, t)| + \|\boldsymbol{f}_2(\boldsymbol{x}, t) - \boldsymbol{h}_2(\boldsymbol{x}, t)\|\right\}$$

$$\quad + C_1 |m_t'| \{C_5 |p_t(\boldsymbol{x}) - f_1(\boldsymbol{x}, t)| + \|\boldsymbol{f}_3(\boldsymbol{x}, t) - \boldsymbol{h}_3(\boldsymbol{x}, t)\|\}$$

$$\leq \tilde{C} \left\{(|\sigma_t'| \sqrt{\log N} + |m_t'|) |f_1(\boldsymbol{x}, t) - p_t(\boldsymbol{x})| + |\sigma_t'| \|\boldsymbol{f}_2(\boldsymbol{x}, t) - \boldsymbol{h}_2(\boldsymbol{x}, t)\| + |m_t'| \|\boldsymbol{f}_3(\boldsymbol{x}, t) - \boldsymbol{h}_3(\boldsymbol{x}, t)\|\right\}. \tag{54}$$

We evaluate the integral on $\{\boldsymbol{x} \mid \|\boldsymbol{x}\|_\infty \leq m_t\}$. From the bound (54), we have

$$I_{B,1} := \int_{\|\boldsymbol{x}\|_\infty \leq m_t} \mathbf{1}[p_t(\boldsymbol{x}) \geq N^{-\frac{2s+\omega}{d}}] \|\boldsymbol{f}_4(\boldsymbol{x}, t) - \boldsymbol{v}_t(\boldsymbol{x})\|^2 p_t(\boldsymbol{x}) d\boldsymbol{x}$$

$$\leq C' \left[\{(\sigma_t')^2 \log N + (m_t')^2\} \int_{\|\boldsymbol{x}\|_\infty \leq m_t} \mathbf{1}[p_t(\boldsymbol{x}) \geq N^{-\frac{2s+\omega}{d}}] |f_1(\boldsymbol{x}, t) - p_t(\boldsymbol{x})|^2 p_t(\boldsymbol{x}) d\boldsymbol{x}\right.$$

$$\quad + (\sigma_t')^2 \int_{\|\boldsymbol{x}\|_\infty \leq m_t} \mathbf{1}[p_t(\boldsymbol{x}) \geq N^{-\frac{2s+\omega}{d}}] \|\boldsymbol{f}_2(\boldsymbol{x}, t) - \boldsymbol{h}_2(\boldsymbol{x}, t)\|^2 p_t(\boldsymbol{x}) d\boldsymbol{x}$$

$$\quad \left. + (m_t')^2 \int_{\|\boldsymbol{x}\|_\infty \leq m_t} \mathbf{1}[p_t(\boldsymbol{x}) \geq N^{-\frac{2s+\omega}{d}}] \|\boldsymbol{f}_3(\boldsymbol{x}, t) - \boldsymbol{h}_3(\boldsymbol{x}, t)\|^2 p_t(\boldsymbol{x}) d\boldsymbol{x}\right]. \tag{55}$$

We write $J_B^{(1)}$, $J_B^{(2)}$, and $J_B^{(3)}$ for the three integrals that appear in the right-hand side of (55).

We will show only the derivation of an upper bound for $J_B^{(2)}$, since the other two cases are similar. Recall that by the definition of $\boldsymbol{f}_2$ and $\boldsymbol{h}_2$,

$$\boldsymbol{f}_2(\boldsymbol{x}, t) - \boldsymbol{h}_2(\boldsymbol{x}, t) = \int_{I^d} \frac{\boldsymbol{x} - m_t \boldsymbol{y}}{\sigma_t} \frac{1}{(\sqrt{2\pi}\sigma_t)^d} e^{-\frac{\|\boldsymbol{x} - m_t \boldsymbol{y}\|^2}{2\sigma_t^2}} (f_N(\boldsymbol{y}) - p_0(\boldsymbol{y})) d\boldsymbol{y}.$$

Then, using $p_t(x) \leq C_1$, we have

$$J_B^{(2)} \leq C_1 \int_{\|\boldsymbol{x}\|_\infty \leq m_t} \mathbf{1}[p_t(\boldsymbol{x}) \geq N^{-\frac{2s+\omega}{d}}] \left\| \int_{I^d} \frac{\boldsymbol{x} - m_t \boldsymbol{y}}{\sigma_t} \frac{1}{(\sqrt{2\pi}\sigma_t)^d} e^{-\frac{\|\boldsymbol{x} - m_t \boldsymbol{y}\|^2}{2\sigma_t^2}} (f_N(\boldsymbol{y}) - p_0(\boldsymbol{y})) d\boldsymbol{y} \right\|^2 d\boldsymbol{x}$$

$$\leq C_1 \int_{\|\boldsymbol{x}\|_\infty \leq m_t} \left\| \frac{1}{m_t^d} \int_{\mathbb{R}^d} \mathbf{1}[\|\boldsymbol{y}\|_\infty \leq 1] \frac{\boldsymbol{x} - m_t \boldsymbol{y}}{\sigma_t} \left(\frac{m_t}{\sqrt{2\pi}\sigma_t}\right)^d e^{-\frac{\|\boldsymbol{y} - \boldsymbol{x}/m_t\|^2}{2(\sigma_t/m_t)^2}} (f_N(\boldsymbol{y}) - p_0(\boldsymbol{y})) d\boldsymbol{y} \right\|^2 d\boldsymbol{x}$$

$$\leq \frac{C_1}{m_t^{2d}} \int_{\|\boldsymbol{x}\|_\infty \leq m_t} \int_{\mathbb{R}^d} \mathbf{1}[\|\boldsymbol{y}\|_\infty \leq 1] \left\| \frac{\boldsymbol{x} - m_t \boldsymbol{y}}{\sigma_t} \right\|^2 \left(\frac{m_t}{\sqrt{2\pi}\sigma_t}\right)^d e^{-\frac{\|\boldsymbol{y} - \boldsymbol{x}/m_t\|^2}{2(\sigma_t/m_t)^2}} (f_N(\boldsymbol{y}) - p_0(\boldsymbol{y}))^2 d\boldsymbol{y} d\boldsymbol{x}$$

$$= \frac{C_1}{m_t^d} \int_{\|\boldsymbol{x}\|_\infty \leq m_t} \int_{\mathbb{R}^d} \mathbf{1}[\|\boldsymbol{y}\|_\infty \leq 1] \left\| \frac{\boldsymbol{x} - m_t \boldsymbol{y}}{\sigma_t} \right\|^2 \frac{1}{(\sqrt{2\pi}\sigma_t)^d} e^{-\frac{\|\boldsymbol{x} - m_t \boldsymbol{y}\|^2}{2\sigma_t^2}} (f_N(\boldsymbol{y}) - p_0(\boldsymbol{y}))^2 d\boldsymbol{y} d\boldsymbol{x},$$

$$\tag{56}$$

where the third line uses Jensen's inequality for $\|\cdot\|^2$. For $t \in 3N^{-\frac{\kappa^{-1}-\delta}{d}}$ with sufficiently large $N$, we can find $c_0 > 0$ such that $m_t \geq c_0$ on the time interval $[T_0, 3N^{-\frac{\kappa^{-1}-\delta}{d}}]$. We can thus further obtain for some $C' > 0$

$$J_B^{(2)} \leq C' \int_{I^d} \int_{\mathbb{R}^d} \frac{\|\boldsymbol{x} - m_t \boldsymbol{y}\|^2}{\sigma_t^2} \frac{1}{(\sqrt{2\pi}\sigma_t)^d} e^{-\frac{\|\boldsymbol{x} - m_t \boldsymbol{y}\|^2}{2\sigma_t^2}} d\boldsymbol{x} (f_N(\boldsymbol{y}) - p_0(\boldsymbol{y}))^2 d\boldsymbol{y}$$

$$= dC' \int_{I^d} (f_N(\boldsymbol{y}) - p_0(\boldsymbol{y}))^2 d\boldsymbol{y}$$

$$= dC' \|f_N - p_0\|_{L^2(I^d)}^2$$

$$\leq C'' N^{-\frac{2s}{d}} \tag{57}$$

by the choice of $f_N$. Similarly, we can prove that $J_B^{(1)}$ and $J_B^{(3)}$ have the same upper bounds of $N^{-\frac{2s}{d}}$ order. This proves that there is $C_{B,1} > 0$ such that

$$I_{B,1} \leq C_{B,1} \left\{ (\sigma_t')^2 \log N + (m_t')^2 \right\} N^{-\frac{2s}{d}}. \tag{58}$$

**(VI-b) case:** $m_t \leq \|\boldsymbol{x}\|_\infty \leq m_t + C_4 \sigma_t \sqrt{\log N}$.

Unlike case (i), we do not have a constant lower bound of $p_t(\boldsymbol{x})$ in this region, and thus we resort to the bound $p_t(\boldsymbol{x}) \geq N^{-(2s+\omega)/d}$, that is $1/p_t(\boldsymbol{x}) \leq N^{(2s+\omega)/d}$ and $1/f_1(\boldsymbol{x}, t) \leq N^{(2s+\omega)/d}$. We have

$$\left\| \frac{\sigma_t' \boldsymbol{f}_2(\boldsymbol{x}, t) + m_t' \boldsymbol{f}_3(\boldsymbol{x}, t)}{f_1(\boldsymbol{x})} - \frac{\sigma_t' \boldsymbol{h}_2(\boldsymbol{x}, t) + m_t' \boldsymbol{h}_3(\boldsymbol{x}, t)}{p_t(\boldsymbol{x})} \right\|$$

$$\leq \frac{1}{f_1(\boldsymbol{x}, t)} \|(\sigma_t' \boldsymbol{f}_2(\boldsymbol{x}, t) + m_t' \boldsymbol{f}_2(\boldsymbol{x}, t)) - (\sigma_t' \boldsymbol{h}_2(\boldsymbol{x}, t) + m_t' \boldsymbol{h}_3(\boldsymbol{x}, t))\|$$

$$\quad + \|\boldsymbol{v}_t(\boldsymbol{x})\| \frac{1}{f_1(\boldsymbol{x}, t)} |f_1(\boldsymbol{x}) - p_t(\boldsymbol{x})|$$

$$\leq N^{(2s+\omega)/d} \tilde{C} \Big\{ (|\sigma_t'| \sqrt{\log N} + |m_t'|) |p_t(\boldsymbol{x}) - f_1(\boldsymbol{x}, t)| + |\sigma_t'| \|\boldsymbol{f}_2(\boldsymbol{x}, t) - \boldsymbol{h}_2(\boldsymbol{x}, t)\|$$

$$\quad + |m_t'| \|\boldsymbol{f}_3(\boldsymbol{x}, t) - \boldsymbol{h}_3(\boldsymbol{x}, t)\| \Big\}, \tag{59}$$

where in the last inequality we use Lemma 19 (i).

Let $\Delta_{t,N} := \{\boldsymbol{x} \in \mathbb{R}^d \mid m_t \leq \|\boldsymbol{x}\|_\infty \leq m_t + C_4 \sigma_t \sqrt{\log N}\}$. By the same argument using Jensen's inequality as the derivation of (56), we can obtain

$$
I_{B,2} := \int_{\Delta_{t,N}} \mathbf{1}[p_t(\boldsymbol{x}) \geq N^{-\frac{2s+\omega}{d}}] \|\boldsymbol{f}_4(\boldsymbol{x},t) - \boldsymbol{v}_t(\boldsymbol{x})\|^2 p_t(\boldsymbol{x}) d\boldsymbol{x}
$$

$$
\leq C''' N^{\frac{4s+2\omega}{d}} \left[ \{(\sigma_t')^2 \log N + (m_t')^2\} \int_{\Delta_{t,N}} \int_{I^d} \frac{1}{(\sqrt{2\pi}\sigma_t)^d} e^{-\frac{\|\boldsymbol{x} - m_t \boldsymbol{y}\|^2}{2\sigma_t^2}} (f_N(\boldsymbol{y}) - p_0(\boldsymbol{y}))^2 d\boldsymbol{y} d\boldsymbol{x} \right.
$$

$$
+ (\sigma_t')^2 \int_{\Delta_{t,N}} \int_{I^d} \left\| \frac{\boldsymbol{x} - m_t \boldsymbol{y}}{\sigma_t} \right\|^2 \frac{1}{(\sqrt{2\pi}\sigma_t)^d} e^{-\frac{\|\boldsymbol{x} - m_t \boldsymbol{y}\|^2}{2\sigma_t^2}} (f_N(\boldsymbol{y}) - p_0(\boldsymbol{y}))^2 d\boldsymbol{y} d\boldsymbol{x}
$$

$$
\left. + (m_t')^2 \int_{\Delta_{t,N}} \int_{I^d} \|\boldsymbol{y}\|^2 \frac{1}{(\sqrt{2\pi}\sigma_t)^d} e^{-\frac{\|\boldsymbol{x} - m_t \boldsymbol{y}\|^2}{2\sigma_t^2}} (f_N(\boldsymbol{y}) - p_0(\boldsymbol{y}))^2 d\boldsymbol{y} d\boldsymbol{x} \right] \tag{60}
$$

Due to the factor $N^{(4s+2\omega)/d}$, the integrals must have orders smaller than in the case of (56) to derive the desired bound of $I_{B,2}$. We will make use of Assumption (A1) about the higher-order smoothness around the boundary of $I^d$.

Because the three integrals can be bounded in a similar manner, we focus only on the second one, denoted by $K_B^{(2)}$. Since $\delta, \omega > 0$ can be taken arbitrarily small, we can assume $\check{s} > 6s - 1 + \delta\kappa + 2\omega$. From Lemma 15 with $\varepsilon = N^{-\check{s}/d}$, there is $C_b > 0$, which is independent of $x$, $t$, and sufficiently large $N$, such that

$$
\left| \int_{I^d} \left\| \frac{\boldsymbol{x} - m_t \boldsymbol{y}}{\sigma_t} \right\|^2 \frac{1}{(\sqrt{2\pi}\sigma_t)^d} e^{-\frac{\|\boldsymbol{x} - m_t \boldsymbol{y}\|^2}{2\sigma_t^2}} (f_N(\boldsymbol{y}) - p_0(\boldsymbol{y}))^2 d\boldsymbol{y} \right.
$$

$$
\left. - \int_{A_{\boldsymbol{x}}} \left\| \frac{\boldsymbol{x} - m_t \boldsymbol{y}}{\sigma_t} \right\|^2 \frac{1}{(\sqrt{2\pi}\sigma_t)^d} e^{-\frac{\|\boldsymbol{x} - m_t \boldsymbol{y}\|^2}{2\sigma_t^2}} (f_N(\boldsymbol{y}) - p_0(\boldsymbol{y}))^2 d\boldsymbol{y} \right| \leq N^{-\frac{\check{s}}{d}}, \tag{61}
$$

where $A_{\boldsymbol{x}}$ is given by

$$
A_{\boldsymbol{x}} := \left\{ \boldsymbol{y} \in I^d \mid \left\| \boldsymbol{y} - \frac{\boldsymbol{x}}{m_t} \right\|_\infty \leq C_b \frac{\sigma_t \sqrt{\log N}}{m_t} \right\}.
$$

Note that if $\boldsymbol{x} \in \Delta_{t,N}$ and $\boldsymbol{y} \in A_{\boldsymbol{x}}$, then

$$
-1 \leq y_j \leq -1 + \frac{C_b \sigma_t \sqrt{\log N}}{m_t} \quad \text{or} \quad 1 - \frac{C_b \sigma_t \sqrt{\log N}}{m_t} \leq y_j \leq 1
$$

for each $j = 1, \ldots, d$. Because we assume that $t \leq 3N^{-\frac{\kappa^{-1}-\delta}{d}}$ and $\sigma_t \sim b_0 t^\kappa$, we can assume that $m_t \geq \sqrt{D_0}/2$ from Assumption (A3). Then, for sufficiently large $N$, we see that $\boldsymbol{y} \in I^d \backslash I_N^d$. This can be seen in $\sigma_t \sim b_0 t^\kappa \leq b_0 3^\kappa N^{-\frac{1-\delta\kappa}{d}}$ and thus $C_b \sigma_t \sqrt{\log N}/m_t \leq \frac{2C_b b_0 3^\kappa}{\sqrt{D_0}} N^{-\frac{1-\delta\kappa}{d}}$. For $\boldsymbol{y} \in I^d \backslash I_N^d$, we can use the second bound in (25); $\|f_N - p_0\|_{L^2(I^d \backslash I_N^d)} \leq N^{-\check{s}/d}$. It follows from (61) that

$$
K_B^{(2)} = \int_{\Delta_{t,N}} \int_{I^d} \left\| \frac{\boldsymbol{x} - m_t \boldsymbol{y}}{\sigma_t} \right\|^2 \frac{1}{(\sqrt{2\pi}\sigma_t)^d} e^{-\frac{\|\boldsymbol{x} - m_t \boldsymbol{y}\|^2}{2\sigma_t^2}} (f_N(\boldsymbol{y}) - p_0(\boldsymbol{y}))^2 d\boldsymbol{y} d\boldsymbol{x}
$$

$$
\leq \int_{\Delta_{t,N}} \left\{ \int_{A_{\boldsymbol{x}}} \left\| \frac{\boldsymbol{x} - m_t \boldsymbol{y}}{\sigma_t} \right\|^2 \frac{1}{(\sqrt{2\pi}\sigma_t)^d} e^{-\frac{\|\boldsymbol{x} - m_t \boldsymbol{y}\|^2}{2\sigma_t^2}} (f_N(\boldsymbol{y}) - p_0(\boldsymbol{y}))^2 d\boldsymbol{y} + N^{-\check{s}/d} \right\} d\boldsymbol{x}
$$

$$
\leq \int_{I^d \backslash I_N^d} \int_{\mathbb{R}^d} \left\| \frac{\boldsymbol{x} - m_t \boldsymbol{y}}{\sigma_t} \right\|^2 \frac{1}{(\sqrt{2\pi}\sigma_t)^d} e^{-\frac{\|\boldsymbol{x} - m_t \boldsymbol{y}\|^2}{2\sigma_t^2}} d\boldsymbol{x} (f_N(\boldsymbol{y}) - p_0(\boldsymbol{y}))^2 d\boldsymbol{y} + N^{-\check{s}/d} |\Delta_{t,N}|.
$$

Since the volume $|\Delta_{t,N}|$ is upper bounded by $D' \sigma_t \sqrt{\log N}$ with some constant $D' > 0$, we have

$$
K_B^{(2)} \leq d\|f_N - p_0\|_{L^2(I^d \backslash I_N^d)}^2 + C'(\sigma_t \sqrt{\log N}) N^{-\check{s}/d}
$$

$$
\leq C'' \left( N^{-2\check{s}/d} + N^{-(\check{s}+1-\delta\kappa)/d} \log^{d/2} N \right)
$$

$$
= O\left( N^{-\frac{\check{s}+1-\delta\kappa}{d}} \log^{d/2} N \right), \tag{62}
$$

where the last line uses $\check{s} > 1$ (A1). The integrals $K_B^{(1)}$ and $K_B^{(3)}$ have an upper bound of the same order. As a result, there is $C_{B,2} > 0$ which does not depend on $n$ or $t$ such that

$$I_{B,2} \leq C_{B,2} \left\{ (\sigma_t')^2 \log N + (m_t')^2 \right\} N^{-\frac{\check{s}+1-4s-2\omega-\delta\kappa}{d}}.$$

Since we have taken $\check{s}$ so that $\check{s} > 6s - 1 + \delta\kappa + 2\omega$, we have

$$I_{B,2} \leq C_{B,2} \left\{ (\sigma_t')^2 \log N + (m_t')^2 \right\} N^{-\frac{2s}{d}}. \tag{63}$$

**(VI-c)**

It follows from (58) and (63) that there is $C_B > 0$ such that

$$I_B \leq C_B \left\{ (\sigma_t')^2 \log N + (m_t')^2 \right\} N^{-\frac{2s}{d}} \tag{64}$$

for sufficiently large $N$.

**(V) Concluding the proof.**

Combining (41), (46), (50), and (64) leads to the upper bound of the statement of the theorem. The argument of the network size in part (III) proves the corresponding statement. $\qquad\square$

## C.5 Bounds of the approximation error for larger $t$

This subsection gives a proof of Theorem 8. The proof is parallel to that of Theorem 7 in many places, while the smoothness of the target density is more helpful.

**(I)' Restriction of the integral.** In a similar manner to part (I) of Section C.4, there is $C_8 > 0$, which does not depend on $t$ such that for any neural network $\phi(\boldsymbol{x}, t)$ with $\|\phi(\boldsymbol{x}, t)\| \leq C_3\{|\sigma_t'|\sqrt{\log N} + |m_t'|\}$ the bound

$$\int_{\|x\| \geq m_t + C_8\sqrt{\log N}} p_t(\boldsymbol{x})\|\phi(\boldsymbol{x}, t) - \boldsymbol{v}_t(\boldsymbol{x})\|^2 d\boldsymbol{x} \lesssim \{|\sigma_t'|\sqrt{\log N} + |m_t'|\}N^{-\eta}$$

holds for any $t \in [N^{-\frac{\kappa^{-1}-\delta}{d}}, 1]$. Also, in a similar way, we can restrict the integral to the region $\{\boldsymbol{x} \mid p_t(\boldsymbol{x}) \geq N^{-\eta}\}$ up to a negligible difference. Consequently, we obtain

$$\int_{\mathbb{R}^d} p_t(\boldsymbol{x})\|\phi(\boldsymbol{x}, t) - \boldsymbol{v}_t(\boldsymbol{x})\|^2 d\boldsymbol{x}$$
$$= \int_{\|x\| \leq m_t + C_8\sqrt{\log N}} \mathbf{1}[p_t(\boldsymbol{x}) \geq N^{-\eta}]p_t(\boldsymbol{x})\|\phi(\boldsymbol{x}, t) - \boldsymbol{v}_t(\boldsymbol{x})\|^2 d\boldsymbol{x}$$
$$+ O\left(\{|\sigma_t'|\sqrt{\log N} + |m_t'|\}N^{-\eta}\right). \tag{65}$$

**(II)' Decomposition of integral.** We consider a $B$-spline approximation of $p_t(\boldsymbol{x})$. Unlike Section C.4, we can regard $p_{t_*}$ as the target distribution, which is of class $C^\infty$. This will cause a tighter bound than in Section C.4 and easier analysis. More precisely, it is easy to see that $p_t$, the convolution between $p_0$ and the Gaussian distribution $N_d(m_t\boldsymbol{y}, \sigma_t^2 I_d)$, can be rewritten as the convolution between $p_{t_*}$ and $N_d(\tilde{m}_t\boldsymbol{y}, \tilde{\sigma}_t^2 I_d)$ for any $t > t_*$, that is,

$$p_t(x) = \int_{\mathbb{R}^d} \frac{1}{(\sqrt{2\pi}\tilde{\sigma}_t)^d} e^{-\frac{\|x - \tilde{m}_t y\|^2}{2\tilde{\sigma}_t^2}} p_{t_*}(y) dy,$$

where

$$\tilde{m}_t := \frac{m_t}{m_{t_*}}, \qquad \tilde{\sigma}_t := \sqrt{\sigma_t^2 - \left(\frac{m_t}{m_{t_*}}\right)^2 \sigma_{t_*}^2}. \tag{66}$$

We can thus apply a similar argument to Section C.4.

We use a $B$-spline approximation of $p_{t_*}$. For $\eta > 0$, take $\alpha \in \mathbb{N}$ such that $\alpha > \frac{3d\eta}{2\delta\kappa}$. It is easy to see that the derivatives of $p_{t_*}(\boldsymbol{x})$ satisfy

$$\left\| \frac{\partial^k}{\partial x_{i_1} \cdots \partial x_{i_k}} p_{t*}(\boldsymbol{x}) \right\| \leq \frac{C_a}{\sigma_{t_*}^k}$$

for any $k \leq \alpha$ and $(i_1, \ldots, i_k)$. From Assumption (A3), there are $t^\dagger \in [0, 1]$ and $b^\dagger > 0$ such that $\sigma_t \geq b^\dagger t^\kappa$ for any $0 \leq t \leq t^\dagger$. If we set $c^\dagger := (t^\dagger)^\kappa > 0$, then $\sigma_t \geq b^\dagger t^\kappa \vee c^\dagger$ for any $t \in [0, 1]$. We can see that

$$\frac{p_{t_*}}{t_*^{-\alpha\kappa} \vee c^\dagger} \in B^\alpha_{\infty,\infty}(\mathbb{R}^d)$$

holds, because for any $k \leq \alpha$ we have

$$\left\| \frac{\partial^k}{\partial x_{i_1} \cdots \partial x_{i_k}} \frac{p_{t_*}(\boldsymbol{x})}{t_*^{-\alpha\kappa} \vee c^\dagger} \right\| \leq \frac{C_\alpha(b^\dagger t_*^{-k\kappa} \wedge (c^\dagger)^{-k})}{t_*^{-\alpha\kappa} \vee c^\dagger} \leq C_\alpha(b^\dagger t_*^{(\alpha-k)\kappa} \wedge (c^\dagger)^{-(k+1)}) \leq C_\alpha((b^\dagger \wedge (c^\dagger)^{-(k+1)}),$$

which implies $\frac{p_{t_*}}{t_*^{-\alpha\kappa}\vee c^\dagger} \in W^\alpha_\infty(\mathbb{R}^d)$ and $\|\frac{p_{t_*}}{t_*^{-\alpha\kappa}\vee c^\dagger}\|_{W^\alpha_\infty(\mathbb{R}^d)} \leq C_\alpha((b^\dagger \wedge (c^\dagger)^{-(k+1)})$ (constant).

Notice that, by a similar argument as in the proof of Lemma 19 (ii), there is $C_5 > 0$ such that

$$\int_{\|y\|_\infty \geq C_5\sqrt{\log N}} (\|\boldsymbol{y}\|^2 + 1) p_{t_*}(\boldsymbol{y}) d\boldsymbol{y} \leq N^{-3\eta} \tag{67}$$

holds. We therefore consider a $B$-spline approximation on $[-C_5\sqrt{\log N}, C_5\sqrt{\log N}]^d$. Letting

$$N_* := \lceil t_*^{-d\kappa} N^{\delta\kappa} \rceil$$

be the number of $B$-spline bases, from Theorem 12, we can find a function $f_{N^*}$ of the form

$$f_{N^*}(\boldsymbol{x}) = (t_*^{-\alpha\kappa} \vee c^\dagger) \sum_{i=1}^{N^*} \alpha_i \mathbf{1}[\|\boldsymbol{x}\|_\infty \leq C_5\sqrt{\log N}] M^d_{k_i, j_i}(\boldsymbol{x})$$

with $|\alpha_i| \leq 1$ and $C_9 > 0$ such that

$$\|p_{t_*} - f_{N^*}\|_{L^2([-C_5\sqrt{\log N}, C_5\sqrt{\log N}]^d)} \leq C_9(\log N)^{\alpha/2}(N^*)^{-\frac{\alpha}{d}}(t_*^{-\alpha\kappa} \vee c^\dagger)$$

holds.

From $N^* \geq t_*^{-d\kappa} N^{\delta\kappa}$ and $\alpha > \frac{3d\eta}{2\delta\kappa}$, the right-hand side is bounded by

$$C'(\log N)^{\alpha/2} N^{-\delta\alpha\kappa/d} \leq C'N^{-3\eta/2}$$

for sufficiently large $N$, which implies

$$\|p_{t_*} - f_{N^*}\|_{L^2([-C_5\sqrt{\log N}, C_5\sqrt{\log N}]^d)} \leq C_{10} N^{-3\eta/2} \tag{68}$$

for sufficiently large $N$. Note also that the coefficient $\tilde{\alpha}_i := \alpha_i(t_*^{-\alpha\kappa} \vee c^\dagger)$ of the basis $M^d_{k_i, j_i}$ in $f_{N^*}$ is bounded by

$$|\tilde{\alpha}_i| \leq t_*^{-\alpha\kappa} \vee c^\dagger \leq N^{\frac{\kappa^{-1}-\delta}{d}\alpha\kappa} = N^{\frac{\alpha(1-\delta\kappa)}{d}}.$$

In a similar manner to part (II) of Section C.4, define $f_1, \boldsymbol{f}_2, \boldsymbol{f}_3$, and $\boldsymbol{f}_4$ using $f_{N^*}$ as

$$f_1(\boldsymbol{x}, t) := \tilde{f}_1(\boldsymbol{x}, t) \vee N^{-\eta} \quad \text{with} \quad \tilde{f}_1(\boldsymbol{x}, t) := \int \frac{1}{(\sqrt{2\pi}\tilde{\sigma}_t)^d} e^{-\frac{\|\boldsymbol{x} - \tilde{m}_t \boldsymbol{y}\|^2}{2\tilde{\sigma}_t^2}} f_{N^*}(\boldsymbol{y}) d\boldsymbol{y}, \tag{69}$$

$$\boldsymbol{f}_2(\boldsymbol{x}, t) := \int \frac{\boldsymbol{x} - \tilde{m}_t \boldsymbol{y}}{\tilde{\sigma}_t} \frac{1}{(\sqrt{2\pi}\tilde{\sigma}_t)^d} e^{-\frac{\|\boldsymbol{x} - \tilde{m}_t \boldsymbol{y}\|^2}{2\tilde{\sigma}_t^2}} f_{N^*}(\boldsymbol{y}) d\boldsymbol{y}.$$

$$\boldsymbol{f}_3(\boldsymbol{x}, t) := \int \boldsymbol{y} \frac{1}{(\sqrt{2\pi}\tilde{\sigma}_t)^d} e^{-\frac{\|\boldsymbol{x} - \tilde{m}_t \boldsymbol{y}\|^2}{2\tilde{\sigma}_t^2}} f_{N^*}(\boldsymbol{y}) d\boldsymbol{y}, \tag{70}$$

$$\boldsymbol{f}_4(\boldsymbol{x}, t) := \frac{\tilde{\sigma}_t' \boldsymbol{f}_2(\boldsymbol{x}, t) + \tilde{m}_t' \boldsymbol{f}_3(\boldsymbol{x}, t)}{f_1(\boldsymbol{x}, t)} \mathbf{1}\left[\left|\frac{\boldsymbol{f}_2}{f_1}\right| \leq C_5\sqrt{\log N}\right] \mathbf{1}\left[\left|\frac{\boldsymbol{f}_3}{f_1}\right| \leq C_5\right].$$

We have a similar decomposition of the integral to (46):

$$\int_{D_{t,N}} \mathbf{1}[p_t(\boldsymbol{x}) \geq N^{-\eta}] \|\phi(\boldsymbol{x}, t) - \boldsymbol{v}_t(\boldsymbol{x})\|^2 p_t(\boldsymbol{x}) d\boldsymbol{x}$$

$$\leq \int_{D_{t,N}} \mathbf{1}[p_t(\boldsymbol{x}) \geq N^{-\eta}] \|\phi(\boldsymbol{x}, t) - \boldsymbol{f}_4(\boldsymbol{x}, t)\|^2 p_t(\boldsymbol{x}) d\boldsymbol{x}$$

$$+ \int_{D_{t,N}} \mathbf{1}[p_t(\boldsymbol{x}) \geq N^{-\eta}] \|\boldsymbol{f}_4(\boldsymbol{x}, t) - \boldsymbol{v}_t(\boldsymbol{x})\|^2 p_t(\boldsymbol{x}) d\boldsymbol{x}$$

$$=: \tilde{I}_A + \tilde{I}_B \tag{71}$$

**(III)' Bound of $\tilde{I}_A$ (neural network approximation of B-spline)**

Using exactly the same argument as in part (III) of Section C.4, we can show

$$\tilde{I}_A = O(\text{poly}(\log N)N^{-\eta'})$$

for arbitrary $\eta' > 0$, and thus it is negligible.

The size of the neural network is given by $L = O(\log^4 N)$, $\|W\|_\infty = O(N)$, $S = O(N^*) = O(t_*^{-d\kappa}N^{\delta\kappa})$, and $B = \exp(O(\log N \log \log N)$.

**(IV)' Bound of $\tilde{I}_B$ (B-spline approximation of the true vector field)**

By replacing $p_0$ with $p_{t_*}$ in (52), define $\boldsymbol{h}_2(\boldsymbol{x},t)$ and $\boldsymbol{h}_3(\boldsymbol{x},t)$ with $\tilde{m}_t$ and $\tilde{\sigma}_t$ by

$$\boldsymbol{h}_2(\boldsymbol{x},t) := \int_{\mathbb{R}^d} \frac{\boldsymbol{x} - \tilde{m}_t \boldsymbol{y}}{\tilde{\sigma}_t} \frac{1}{(\sqrt{2\pi}\tilde{\sigma}_t)^d} e^{-\frac{\|\boldsymbol{x} - \tilde{m}_t \boldsymbol{y}\|^2}{2\tilde{\sigma}_t^2}} p_{t_*}(\boldsymbol{y})d\boldsymbol{y},$$

$$\boldsymbol{h}_3(\boldsymbol{x},t) := \int_{\mathbb{R}^d} \boldsymbol{y} \frac{1}{(\sqrt{2\pi}\tilde{\sigma}_t)^d} e^{-\frac{\|\boldsymbol{x} - \tilde{m}_t \boldsymbol{y}\|^2}{2\tilde{\sigma}_t^2}} p_{t_*}(\boldsymbol{y})d\boldsymbol{y}.$$

Then, by a similar argument to (19), we have

$$\|\boldsymbol{f}_4(\boldsymbol{x},t) - \boldsymbol{v}_t(\boldsymbol{x})\| \leq N^\eta \tilde{C} \Big\{ \big(|\tilde{\sigma}_t'|\sqrt{\log N} + |\tilde{m}_t'|\big)|p_t(\boldsymbol{x}) - f_1(\boldsymbol{x},t)| $$
$$+ |\tilde{\sigma}_t'|\|\boldsymbol{f}_2(\boldsymbol{x},t) - \boldsymbol{h}_2(\boldsymbol{x},t)\| + |\tilde{m}_t'|\|\boldsymbol{f}_3(\boldsymbol{x},t) - \boldsymbol{h}_3(\boldsymbol{x},t)\|\Big\}$$

for some constant $\tilde{C}$, and thus

$$\tilde{I}_B \leq C'N^{2\eta} \Bigg[ \big\{(\tilde{\sigma}_t')^2 \log N + (\tilde{m}_t')^2\big\} \int_{D_{t,N}} \left| \int_{\mathbb{R}^d} \frac{1}{(\sqrt{2\pi}\tilde{\sigma}_t)^d} e^{-\frac{\|\boldsymbol{x} - \tilde{m}_t \boldsymbol{y}\|^2}{2\tilde{\sigma}_t^2}} (f_{N^*}(\boldsymbol{y}) - p_{t_*}(\boldsymbol{y}))d\boldsymbol{y} \right|^2 d\boldsymbol{x}$$

$$+ (\tilde{\sigma}_t')^2 \int_{D_{t,N}} \left\| \int_{\mathbb{R}^d} \frac{\boldsymbol{x} - \tilde{m}_t \boldsymbol{y}}{\tilde{\sigma}_t} \frac{1}{(\sqrt{2\pi}\tilde{\sigma}_t)^d} e^{-\frac{\|\boldsymbol{x} - \tilde{m}_t \boldsymbol{y}\|^2}{2\tilde{\sigma}_t^2}} (f_{N^*}(\boldsymbol{y}) - p_{t_*}(\boldsymbol{y}))d\boldsymbol{y} \right\|^2 d\boldsymbol{x}$$

$$+ (\tilde{m}_t')^2 \int_{D_{t,N}} \left\| \int_{\mathbb{R}^d} \boldsymbol{y} \frac{1}{(\sqrt{2\pi}\tilde{\sigma}_t)^d} e^{-\frac{\|\boldsymbol{x} - \tilde{m}_t \boldsymbol{y}\|^2}{2\tilde{\sigma}_t^2}} (f_{N^*}(\boldsymbol{y}) - p_{t_*}(\boldsymbol{y}))d\boldsymbol{y} \right\|^2 d\boldsymbol{x} \Bigg]. \qquad (72)$$

Here, we show a bound of

$$\tilde{J}_{B,2} := \int_{D_{t,N}} \left\| \int_{\mathbb{R}^d} \frac{\boldsymbol{x} - \tilde{m}_t \boldsymbol{y}}{\tilde{\sigma}_t} \frac{1}{(\sqrt{2\pi}\tilde{\sigma}_t)^d} e^{-\frac{\|\boldsymbol{x} - \tilde{m}_t \boldsymbol{y}\|^2}{2\tilde{\sigma}_t^2}} (f_{N^*}(\boldsymbol{y}) - p_{t_*}(\boldsymbol{y}))d\boldsymbol{y} \right\|^2 d\boldsymbol{x}.$$

The other two integrals can be bounded similarly.

Let $\rho := \frac{1}{\sqrt{2}D_0} > 0$, where $D_0$ is given in Assumption (A3). We derive a bound of $\tilde{J}_{B,2}$ in the two cases of $t$ separately: **(IV-a)'** $\tilde{m}_t \geq \rho$, and **(IV-b)'** $\tilde{m}_t \leq \rho$.

**Case (IV-a)':** $\tilde{m}_t \geq \rho$**.**

By rewriting the inner integral on $\boldsymbol{y}$ by a Gaussian integral, we have

$$
\begin{aligned}
\tilde{J}_{B,2} &= \int_{D_{t,N}} \frac{1}{\tilde{m}_t^{2d}} \left\| \int_{\mathbb{R}^d} \frac{\boldsymbol{x} - \tilde{m}_t \boldsymbol{y}}{\tilde{\sigma}_t} \left( \frac{\tilde{m}_t}{\sqrt{2\pi}\tilde{\sigma}_t} \right)^d e^{-\frac{\tilde{m}_t^2 \|\boldsymbol{y} - \boldsymbol{x}/\tilde{m}_t\|^2}{2\tilde{\sigma}_t^2}} (f_{N^*}(\boldsymbol{y}) - p_{t_*}(\boldsymbol{y})) d\boldsymbol{y} \right\|^2 d\boldsymbol{x} \\
&\leq \int_{D_{t,N}} \frac{1}{\tilde{m}_t^{2d}} \int_{\mathbb{R}^d} \left\| \frac{\boldsymbol{x} - \tilde{m}_t \boldsymbol{y}}{\tilde{\sigma}_t} \right\|^2 \left( \frac{\tilde{m}_t}{\sqrt{2\pi}\tilde{\sigma}_t} \right)^d e^{-\frac{\tilde{m}_t^2 \|\boldsymbol{y} - \boldsymbol{x}/\tilde{m}_t\|^2}{2\tilde{\sigma}_t^2}} (f_{N^*}(\boldsymbol{y}) - p_{t_*}(\boldsymbol{y}))^2 d\boldsymbol{y} d\boldsymbol{x} \\
&\leq \int_{D_{t,N}} \frac{1}{\tilde{m}_t^d} \int_{\mathbb{R}^d} \left\| \frac{\boldsymbol{x} - \tilde{m}_t \boldsymbol{y}}{\tilde{\sigma}_t} \right\|^2 \frac{1}{(\sqrt{2\pi}\tilde{\sigma}_t)^d} e^{-\frac{\|\boldsymbol{x} - \tilde{m}_t \boldsymbol{y}\|^2}{2\tilde{\sigma}_t^2}} (f_{N^*}(\boldsymbol{y}) - p_{t_*}(\boldsymbol{y}))^2 d\boldsymbol{y} d\boldsymbol{x} \\
&\leq (2D_0)^{d/2} \int_{\mathbb{R}^d} \int_{\mathbb{R}^d} \left\| \frac{\boldsymbol{x} - \tilde{m}_t \boldsymbol{y}}{\tilde{\sigma}_t} \right\|^2 \frac{1}{(\sqrt{2\pi}\tilde{\sigma}_t)^d} e^{-\frac{\|\boldsymbol{x} - \tilde{m}_t \boldsymbol{y}\|^2}{2\tilde{\sigma}_t^2}} (f_{N^*}(\boldsymbol{y}) - p_{t_*}(\boldsymbol{y}))^2 d\boldsymbol{x} d\boldsymbol{y} \\
&\leq \rho^{-d/2} d \int_{\mathbb{R}^d} (f_{N^*}(\boldsymbol{y}) - p_{t_*}(\boldsymbol{y}))^2 d\boldsymbol{y} \\
&\leq \rho^{-d/2} d \left[ \int_{[-C_t\sqrt{\log N}, C_5\sqrt{\log N}]^d} (f_{N^*}(\boldsymbol{y}) - p_{t_*}(\boldsymbol{y}))^2 d\boldsymbol{y} + \int_{\|\boldsymbol{y}\| \geq C_5\sqrt{\log N}} p_{t_*}(\boldsymbol{y})^2 d\boldsymbol{y} \right] \\
&\leq \rho^{-d/2} d \left\{ \|f_{N^*} - p_{t_*}\|_{L^2([-C_5\sqrt{\log N}, C_5\sqrt{\log N}]^d)}^2 + N^{-3\eta} \right\} \\
&\leq C N^{-3\eta}
\end{aligned}
$$

where the second line uses Jensen's inequality and the last two lines are based on (67) and (68). The constant $C > 0$ does not depend on $t$ or $N$.

**Case (IV-b)'** $\tilde{m}_t \leq \rho$.

In this case, we can show $\tilde{\sigma}_t^2 \geq \frac{1}{2D_0}$. In fact, from $\tilde{m}_t \leq \rho$, we have

$$
\tilde{\sigma}_t^2 = \sigma_t^2 - \tilde{m}_t^2 \sigma_{t_*}^2 \geq \sigma_t^2 - \rho^2 \sigma_{t_*}^2.
$$

From Assumption (A3), $m_t^2 + \sigma_t^2 \geq D_0^{-1}$. Since $m_t^2 \leq \rho^2 m_{t_*}^2$ by assumption, we have

$$
\sigma_t^2 \geq D_0^{-1} - m_t^2 \geq D_0^{-1} - \rho^2 m_{t_*}^2.
$$

Combining these two inequalities, we obtain

$$
\tilde{\sigma}_t^2 \geq D_0^{-1} - \rho^2(m_{t_*}^2 + \rho_{t_*}^2) \geq D_0^{-1} - \rho^2 D_0 = \frac{1}{2D_0},
$$

where the last equality holds from the definition $\rho = \frac{1}{\sqrt{2D_0}}$.

We divide the integral of $\tilde{I}_{B,2}$ into the regions $\{\boldsymbol{y} \mid \|\boldsymbol{y}\|_\infty \geq C_5\sqrt{\log N}\}$ and $\{\boldsymbol{y} \mid \|\boldsymbol{y}\|_\infty \leq C_5\sqrt{\log N}\}$. In the region $\{\boldsymbol{y} \mid \|\boldsymbol{y}\|_\infty \geq C_5\sqrt{\log N}\}$, using $\tilde{\sigma}_t^2 \geq 1/(2D_0)$ and $f_{N^*}(\boldsymbol{y}) = 0$, we have a bound

$$
\begin{aligned}
&\left\| \int_{\{\|\boldsymbol{y}\|_\infty \geq C_5\sqrt{\log N}\}} \frac{\boldsymbol{x} - \tilde{m}_t \boldsymbol{y}}{\tilde{\sigma}_t} \frac{1}{(\sqrt{2\pi}\tilde{\sigma}_t)^d} e^{-\frac{\|\boldsymbol{x} - \tilde{m}_t \boldsymbol{y}\|^2}{2\tilde{\sigma}_t}} (f_{N^*}(\boldsymbol{y}) - p_{t_*}(\boldsymbol{y})) d\boldsymbol{y} \right\|^2 \\
&\leq \left( \frac{D_0}{\pi} \right)^d (2D_0)^2 \int_{\{\|\boldsymbol{y}\|_\infty \geq C_5\sqrt{\log N}\}} \|\boldsymbol{x} - \tilde{m}_t \boldsymbol{y}\|^2 (p_{t_*}(\boldsymbol{y}))^2 d\boldsymbol{y} \\
&\leq C \left( \frac{D_0}{\pi} \right)^d (2D_0)^2 \int_{\{\|\boldsymbol{y}\|_\infty \geq C_5\sqrt{\log N}\}} \left( C' \log N + \rho^{-2}\|\boldsymbol{y}\|^2 \right) p_{t_*}(\boldsymbol{y}) d\boldsymbol{y} \\
&\leq C'' N^{-3\eta} \log N,
\end{aligned}
$$

where we use (67) and the fact $\boldsymbol{x} \in D_{t,N}$ implies $\|x\|^2 \leq C' \log N$ for some $C'$.

For the other region $\{\boldsymbol{y} \mid \|\boldsymbol{y}\|_\infty \leq C_5\sqrt{\log N}\}$, first notice that for $\boldsymbol{x} \in D_{t,N}$, we have $\|\boldsymbol{x} - \tilde{m}_t\boldsymbol{y}\|/\tilde{\sigma}_t \leq D'\sqrt{\log N}$ for some $D' > 0$. Application of Cauchy-Schwarz inequality derives

$$\left\| \int_{\{\|\boldsymbol{y}\|_\infty \leq C_5\sqrt{\log N}\}} \frac{\boldsymbol{x} - \tilde{m}_t\boldsymbol{y}}{\tilde{\sigma}_t} \frac{1}{(\sqrt{2\pi}\tilde{\sigma}_t)^d} e^{-\frac{\|\boldsymbol{x}-\tilde{m}_t\boldsymbol{y}\|^2}{2\tilde{\sigma}_t^2}} (f_{N^*}(\boldsymbol{y}) - p_{t_*}(\boldsymbol{y}))d\boldsymbol{y} \right\|^2$$

$$\leq D'^2 \log N \left(\frac{D_0}{\pi}\right) \int_{\{\|\boldsymbol{y}\|_\infty \leq C_5\sqrt{\log N}\}} d\boldsymbol{y} \int_{\{\|\boldsymbol{y}\|_\infty \leq C_5\sqrt{\log N}\}} (f_{N^*}(\boldsymbol{y}) - p_{t_*}(\boldsymbol{y}))^2 d\boldsymbol{y}$$

$$\leq D''(\log N)^{\frac{d}{2}+1} \|f_{N^*} - p_{t_*}\|_{L^2([-C_5\sqrt{\log N}^2, C_5\sqrt{\log N}]^d)}$$

$$\leq D''(\log N)^{\frac{d}{2}+1} N^{-3\eta}.$$

From the above two cases (IV-a)' and (IV-b)', we have for any $t \in [t_*, 1]$

$$\tilde{J}_{B,2} \leq \text{poly}(\log N) N^{-\eta}.$$

As a consequence, there is a constant $C''$ that does not depend on $t$ and $m \in \mathbb{N}$ such that

$$\tilde{I}_B \leq C''\{(\sigma_t')^2 \log N + (m_t')^2\} N^{-\eta}\text{poly}(\log N).$$

The factor $\text{poly}(\log N)$ can be erased if we take a larger $\eta$ in the proof. $\qquad\square$

## D   APPROXIMATION OF FUNCTIONAL OPERATIONS BY NEURAL NETWORKS

This section reviews the accuracy of the approximation and the increase in complexity when we approximate functional operations by neural networks. The following results are shown in Oko et al. (2023, Section F) as well and in more original literature Nakada and Imaizumi (2020),Petersen and Voigtlaender (2018),Schmidt-Hieber (2019).

The operations used directly in this article are `recip`, `mult`, `clip`, and `sw`. The usage in Section C.4 (III) is explained as examples.

### D.1   CLIPPING FUNCTION

First, we consider the realization of the component-wise clipping function.

**Lemma 20.** *For any $a, b \in \mathbb{R}^d$ with $a_i \leq b_i$ ($i = 1, 2, \ldots, d$), there exists a neural network* $\text{clip}(\boldsymbol{x}; a, b) \in \mathcal{M}(L, W, S, B)$ *with $L = 2$, $W = (d, 2d, d)^T$, $S = 7d$, and $B = \max_{1 \leq i \leq d} \max\{|a_i|, b_i\}$ such that*

$$\text{clip}(\boldsymbol{x}; a, b)_i = \min\{b_i, \max\{x_i, a_i\}\} \quad (i = 1, 2, \ldots, d)$$

*holds. When $a_i = c_{min}$ and $b_i = c_{max}$ for all $i$, we also use the notation* $\text{clip}(\boldsymbol{x}; c_{min}, c_{max}) := \text{clip}(\boldsymbol{x}; a, b)$.

### D.2   RECIPROCAL FUNCTION

Second, the reciprocal function $x \mapsto 1/x$ is approximated by neural networks as follows.

**Lemma 21.** *For any $0 < \varepsilon < 1$, there is* $\text{recip}(\boldsymbol{x}') \in \mathcal{M}(L, W, S, B)$ *such that*

$$\left| \text{recip}(\boldsymbol{x}') - \frac{1}{x} \right| \leq \varepsilon + \frac{|\boldsymbol{x} - x'|}{\varepsilon^2} \tag{73}$$

*holds for any $x \in [\varepsilon, \varepsilon^{-1}]$ and $x' \in \mathbb{R}$ with $L = O(\log^2(\varepsilon^{-1}))$, $\|W\|_\infty = O(\log^3(\varepsilon^{-1}))$, $S = O(\log^4(\varepsilon^{-1}))$, and $B = O(\varepsilon^{-2})$.*

### D.3   MULTIPLICATION

**Lemma 22.** *Let $d \geq 2$, $C \geq 1$, $0 < \epsilon_{err} \leq 1$. For any $\varepsilon > 0$, there exists a neural network* $\text{mult}(\boldsymbol{x}_1, x_2, \ldots, x_d) \in \mathcal{M}(L, W, S, B)$ *with $L = O(d \log \varepsilon^{-1} + d \log C)$, $\|W\|_\infty = 48d$, $S =*

$O(d \log \varepsilon^{-1} + d \log C))$, $B = C^d$ such that (i)

$$\left| \text{mult}(x_1', \ldots, x_d') - \prod_{d'=1}^{d} x_i' \right| \le \varepsilon + dC^{d-1}\epsilon_{err},$$

holds for all $x \in [-C, C]^d$ and $\boldsymbol{x}' \in R^d$ with $\|\boldsymbol{x} - \boldsymbol{x}'\|_\infty \le \epsilon_{err}$, (ii) $|\text{mult}(\boldsymbol{x})| \le C^d$ for all $x \in \mathbb{R}^d$, and (iii) $\text{mult}(x_1', \ldots, x_d') = 0$ if at least one of $x_i'$ is 0.

We note that some of $x_i, x_j$ $(i \ne j)$ can be shared; for $\prod_{i=1}^{I} x_{\alpha_i}$ with $\alpha_i \in \mathbb{Z}_+$ $(i = 1, \ldots, I)$ and $\sum_{i=1}^{I} \alpha_i = d$, there exists a neural network satisfying the same bounds as above; the network is denoted by $\text{mult}(\boldsymbol{x}; \alpha)$.

## D.4 SWITCHING

**Lemma 23.** *Let* $t_1 < t_2 < s_1 < s_2$, *and* $f(\boldsymbol{x}, t)$ *be a scaler-valued function. Assume that* $|\varphi_1(\boldsymbol{x}, t) - f(\boldsymbol{x}, t)| \le \varepsilon$ *on* $[t_1, s_1]$ *and* $|\varphi_2(\boldsymbol{x}, t) - f(\boldsymbol{x}, t)| \le \varepsilon$ *on* $[t_2, s_2]$. *Then, there exist neural networks* $\text{sw}_1(t; t_2, s_1)$ *and* $\text{sw}_2(t; t_2, s_1)$ *in* $\mathcal{M}(L, W, S, B)$ *with* $L = 3$, $W = (1, 2, 1, 1)^T$, $S = 8$, *and* $B = \max\{t_1, (s_1 - t_2)^{-1}\}$ *such that*

$$|\text{sw}_1(t; t_2, s_1)\varphi_1(x, t) + \text{sw}_2(t; t_2, s_1)\varphi_2(x, t) - f(x, t)| \le \varepsilon$$

*holds for any* $t \in [t_1, s_2]$.

## D.5 CONSTRUCTION OF NETWORK IN SECTION C.4 (III)

We detail the network size required for the approximation procedure presented in Section C.4 (III).

**Clipping to** $\zeta_1$**:** From (39) and assumption (A3), $\phi_5$ can be upper bounded by $N^{K_0+1}$ for sufficiently large $N$. Then, from Lemma 20, we can see that in the clipping of $\phi_5$, the increase in model sizes is constant depending on $d$ except $B$, which is multiplied by the upper bound $N^{K_0+1}$.

**Approximating** $f_1^{-1}$ **by** $\zeta_2$**:** Since we assume $f_1 \ge N^{-(2s+\omega)/d}$, clipping $\phi_5$ from below by $N^{-(2s+\omega)/d}$ does not increase the difference; thus $|\zeta_1 - f_1| \le D_5 N^{-\eta}$. In Lemma 21, substituting $x' = \zeta_1$, $x = f_1$, and $\varepsilon = N^{-\chi}$ for $\chi > \chi_0 + (2s + \omega)/d$ with an arbitrary $\chi_0 > 0$, we have

$$\left| \text{recip}(\zeta_1(\boldsymbol{x}, t)) - \frac{1}{f_1(\boldsymbol{x}, t)} \right| \le N^{-\chi} + N^{2\chi}|\zeta_1(\boldsymbol{x}, t) - f_1(\boldsymbol{x}, t)|.$$

Since $\eta > 0$ is arbitrary, by setting $\eta$ and $\chi$ so that $\eta > 3\chi$, we have

$$\left| \text{recip}(\zeta_1(\boldsymbol{x}, t)) - \frac{1}{f_1(\boldsymbol{x}, t)} \right| \le (D_5 + 1)N^{-\chi}. \tag{74}$$

This is achieved by a neural network with $L = O(\log^2 N)$, $S = O(\log^4 N)$, $\|W\|_\infty = O(\log \log N)$ and $B = O(N^{2\chi})$.

$\zeta_3 = \text{mult}(\zeta_2, \phi_6)$**:** Note that $|\zeta_2| \le N^{(2s+\omega)/d}$, $\|\boldsymbol{f}_2 - \phi_6\| = O(N^{-\eta})$, and $|\zeta_2 - f_1^{-1}| \le O(N^{-\chi})$ from (74). We have also taken $\eta$ so that $\eta > 3\chi$. In applying Lemma 22, we can set $C = N^{(2s+\omega)/d}$ because $|\zeta_2| \le N^{(2s+\omega)/d}$. Also, $\epsilon_{err} := \max\{|\zeta_2 - 1/f_1|, \|\phi_6 - \boldsymbol{f}_2\|\} = \max\{O(N^{-\chi}), O(N^{-\eta})\} = O(N^{-\chi})$. With $d = 2$ and $\varepsilon = N^{-\chi_0}$, we have

$$|\boldsymbol{\zeta}_3(\boldsymbol{x}, t) - \zeta_2 \cdot \phi_6(\boldsymbol{x}, t)| = O(N^{-\chi_0} + 2N^{(2s+\omega)/d}N^{-\chi}) = O(N^{-\chi_0} + 2N^{-\chi_0}) = O(N^{-\chi_0}),$$

where we use the fact that $\chi$ is taken to satisfy $\chi > \chi_0 + (2s + \omega)/d$.

We then obtain

$$\left\| \zeta_3 - \frac{\boldsymbol{f}_2}{f_1} \right\| \le \|\boldsymbol{\zeta}_3 - \phi_6\zeta_2\| + \left\| \phi_6\zeta_2 - \frac{\boldsymbol{f}_2}{f_1} \right\|$$

$$\le \|\boldsymbol{\zeta}_3 - \phi_6\zeta_2\| + \|\phi_6\zeta_2 - \boldsymbol{f}_2\zeta_2\| + \left\| \boldsymbol{f}_2\zeta_2 - \frac{\boldsymbol{f}_2}{f1} \right\|$$

$$= O(N^{-\chi_0}) + O(N^{(2s+\omega)/d}N^{-\eta}) + O(\sqrt{\log N}N^{-\chi})$$

$$= O(N^{-\chi_0}), \tag{75}$$

where in the second last line we use $\|\boldsymbol{f}_2/\boldsymbol{f}_1\| = O(\sqrt{\log N})$ and thus $\|\boldsymbol{f}_2\| = O(\sqrt{\log N})$.

A similar argument derives

$$\left\|\boldsymbol{\zeta}_5 - \frac{\boldsymbol{f}_3}{\boldsymbol{f}_1}\right\| = O(N^{-\chi_0}). \tag{76}$$

For the neural network architecture in this multiplication, $L$ and $S$ are added by the order of $O(\log \varepsilon^{-1} + \log C) = O(\log N)$. The width $W$ is of constant order and thus negligible. The width $W$ is $O(N^{2(2s+\omega)/d})$.

**Clipping to obtain $\boldsymbol{\zeta}_4$ and $\boldsymbol{\zeta}_6$:** For these clipping procedures, the approximating networks have $L$, $W$ and $S$ of constant order, while the weight values $B$ are of $O(\sqrt{\log N})$ and $O(1)$, respectively, and thus they are negligible. The approximation errors are kept as $N^{-\chi_0}$.

**Multiplication to obtain $\boldsymbol{\zeta}_7$ and $\boldsymbol{\zeta}_8$:** As in the previous procedures, clipping by $O(\sqrt{\log N})$ and $O(1)$ does not increase the approximation error, while the increase in the size of the network is negligible.

In a similar manner to Oko et al. (Lemma B.1 2023), we can approximate $\sigma'_t$ and $m'_t$ by neural networks so that $|\sigma'_t - \widehat{\sigma'_t}| = O(N^{-\eta})$ and $|m'_t - \widehat{m'_t}| = O(N^{-\eta})$. The network sizes are $L = O(\log^2 N)$, $\|W\|_\infty = O(\log^2 N)$, $S = O(\log^3 N)$, and $B = O(\log N)$. With arguments similar to those of the previous procedures, we can show

$$\left\|\boldsymbol{\zeta}_7 - (\sigma'_t)\frac{\boldsymbol{f}_2}{\boldsymbol{f}_1}\mathbf{1}\left[\left\|\frac{\boldsymbol{f}_2}{\boldsymbol{f}_1}\right\| \le C_5\sqrt{\log N}\right]\right\| = O(N^{-\chi_0}),$$

and

$$\left\|\boldsymbol{\zeta}_8 - |m'_t|\frac{\boldsymbol{f}_3}{\boldsymbol{f}_1}\mathbf{1}\left[\left\|\frac{\boldsymbol{f}_3}{\boldsymbol{f}_1}\right\| \le C_5\right]\right\| = O(N^{-\chi_0}),$$

In total, we can find a neural network to approximate $\boldsymbol{v}_t(\boldsymbol{x})$ with the approximation error of $O(N^{-\chi_0})$ so that the network has the size of most polynomial orders for $B$ and $\|W\|_\infty$, while $O(\mathrm{poly}(\log N))$ for $S$ and $L$. As a result, the contributions to the log cover number are only $O(\mathrm{poly}(\log N))$.

# E    IDEA OF TIME DIVISION

The basic idea of the time division used to derive the almost optimal minimax convergence rate is depicted in Figure E.1.

$$W_2(\hat{P}_t, P_t) \le \sqrt{t}\left(\int_0^t \int e^{2\int_s^t e^{L_u du}}\|\hat{\boldsymbol{v}}_s(\boldsymbol{x}) - \boldsymbol{v}_s(\boldsymbol{x})\|^2 dP_s(\boldsymbol{x})dxds\right)^{1/2}$$

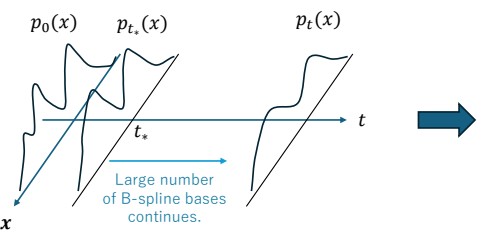
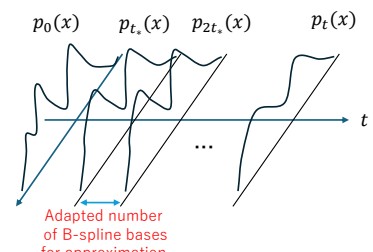

Figure E.1: The idea of time division for deriving the convergence rate

