# OpenReview forum: "Flow matching achieves almost minimax optimal convergence"
_ICLR.cc/2025/Conference — ICLR 2025 Poster_

### Official Review · Reviewer_SoUb · 2024-10-29

**Soundness:** 3
**Presentation:** 3
**Contribution:** 3
**Rating:** 6
**Confidence:** 3

**Summary:**

This paper proves an almost minimax optimality result for a class of flow models. Previously, Oko had shown that diffusion models are minimax optimal under the 1-Wasserstein distance. This paper builds on Oko to show that a class of FMs with terminal Gaussian distribution and paths of the form x_t = \sigma_t x_0 + m_t x_1 (which includes diffusion as a special case) are almost minimax optimal, with a parameter kappa determining the non-optimality. They show this under the p-Wasserstein distance for 1 <= p <= 2.

**Strengths:**

Provides new theoretical results for a family of flow models, showing almost minimax optimality (with 'almost' depending on a specific parameter). The paper is rigorous, clearly written, and clearly places the results in the context of prior work.

**Weaknesses:**

Please see Questions.

**Questions:**

L223: Kappa = 1/2 corresponds to diffusion-FM, correct? So the result actually says that that only diffusion-FM is optimal within the family of FMs you consider? If so, L230 'FM is as strong as diffusion models' seems somewhat misleading -- it seems more like diffusion is actually stronger than FM, except when FM is equivalent to diffusion. Please correct me if I'm wrong; otherwise might want to state this differently.

L225: Can you elaborate on the ways in which your proof technique differs significantly from Oko's? (Since Oko's result is a special case of yours for diffusion-FM and 1-Wasserstein.)

L133: Can anything be said in the more general non-Gaussian case of FMs?

L177: Notation not super clear here. What is P_[1] vs p_[1]?

Theorem 1: It seems like you are further restricting sigma to have the form (1-\tau)^\kappa?

L222: Typo 'revserse'

L224: Diffusion can be expressed as an ODE so I am not sure what you mean here?

---

> ### Author Response · Authors · 2024-11-19
> **Authors' replies**
>
> Thank you for your helpful comments.  We provide our replies below.
>
> Q1 (comparison between FM and DM):  More precisely, Theorem 9 tells that the derived upper bound (almost) attains the universal minimax lower bound at $\kappa=1/2$ in terms of the convergence rate. For other $\kappa > 1/2$, the derived upper bound does not attain the universal lower bound, and there is still a possibility that the convergence rate for $\kappa > 1/2$ can be improved.  To reflect this situation, we would improve the expression as “Both FM and DM can attain the same almost optimal minimax convergence rate for generalization error” (line 230)
>
> Q2 (difference from Oko et al. 2023):  As detailed in response to nHx4 (W1), there are two significant differences from Oko et al. 2023 in terms of proof techniques.
>
> (1) To relate the Wasserstein distance and the squared difference of vector fields (scores in the case of DM), the Alekseev-Grobner Theorem is used for FM, while Girsanov’s Theorem is for DM.  This causes a completely different argument to bound between the final ODE solution and the true probability $W_2(P_0, P_{T_0})$; the analysis of FM requires the uniform Lipschitz constant in Lemma 10, which is based on the form of $\sigma_t$ and $m_t$ considered in this paper and the assumption (A5).
>
> (2) The extension of the paths with $\sigma_t$ and $m_t$  needs more elaborate arguments on the analysis. As a result, we can obtain the influence of the decreasing rate $\kappa$  in $\sigma_t \sim t^\kappa$ on the upper bound of the convergence rate.
>
> Q3: Extension to the non-Gaussian case is an interesting problem, and in fact, we are working on this extension.  In the current proof, special properties of Gaussian distribution help us make bounds at many places.  Mathematically, it is challenging to extend it to general distributions.
>
> Q4: We state at the beginning of Section 2 that we use $P_a$  for probability distribution and $p_a$ for its density function.   We have rewritten the statement of Theorem 1 in the revision.
>
> Q5: This assumption is not so restrictive.  Because $\sigma_t$ is a decreasing function of $t$ and should approach 0 as $t\to 0$, it is natural to set it in the form $t^\kappa$ around 0.   In practice, the OT flow uses $\kappa = 1$ and diffusion flow $\kappa = 1/2$.
>
> Q6 (typo):  We have fixed all the typos in reviewers’ comments.
>
> Q7 (line 224; the relation between FM and DM) See our Global Responses.

---

> > ### Comment · Reviewer_SoUb · 2024-12-02
> >
> > Apologies for the delay. Thank you very much for your response and the answers to my questions. I am happy to keep my initial positive rating.

---

### Official Review · Reviewer_xqeo · 2024-11-02

**Soundness:** 2
**Presentation:** 2
**Contribution:** 2
**Rating:** 6
**Confidence:** 4

**Summary:**

This paper provides near-minimax convergence guarantees for the flow matching (FM) algorithm for $p$-Wasserstein distances. Distinct from diffusion models, FM use ordinary differential equations (ODEs) at inference time instead of stochastic differential equations (SDEs). Their estimator is based on time-partitioned estimators, similar to the analysis of Oko et al (2023). They adopt the estimator for more general parameters (specifically the mean and covariance parameters) to provide an estimator for flow matching.

**Strengths:**

This paper provides (to my understanding) the first estimation rates for flow matching in the context of classical statistical estimation rates. This paper is similar in spirit to the FM paper of Lipman et al (2023), where they use different combinations of mean-variance parameters to define their path. While this work leverages many ideas from Oko et al. (2023), what is especially interesting is this idea that "optimal parameter choices" lead to minimax convergence rates, whereas other choices do not enjoy the same statistical rates.

**Weaknesses:**

The paper could be more clearly written, and the main text is very technical. This paper would benefit substantially from a small figure explaining the construction of the estimator at a high level, and also explaining the reason why the full minimax estimation is not possible. These are overall minor points, but I do believe the paper would benefit greatly from these modifications overall.

**Questions:**

My comments are minor (these are mostly typos I found, but this is far from exhaustive)

1. Line 188 "for probability *density* estimation"?
2. L193 (this might happen in many places): I believe grammatically it makes sense to say "i.id. sample*s*" instead of a single sample
3.  L222: *reverse* not revserve
4. L254: "respectivly" is misspelled
5. This is a question: is there a clean way to track the dependence on the diameter of the set of the support? The assumptions assume the support of the density is in the unit cube. What is the dependence if the radius was arbitrary? This might fall outside the scope of the paper, but I'm curious if the authors have the answer
6. L516: "diffrence" is misspelled

---

> ### Author Response · Authors · 2024-11-19
>
> Weakness:  Thank you for your suggestion to include a figure for explanation.  We agree that a visual explanation could improve accessibility, and we have included a figure to illustrate the basic idea of the time division in Appendix E and mentioned it in line 458 in the revision.
>
> Q1 (probability estimation): FM methods do not estimate a density, but provide samples that approximate the true probability.  We thus used the terminology *probability estimation*.
>
> Q2: (i.i.d. samples)  We agree. We have reflected all of this in the revision.
>
> Q3, 4, 6 (typos):  We have fixed as many typos as possible in addition to those you pointed out.
>
> Q5 (support)  The cube size does not affect the result.  We have mentioned it in line 166 in the revision.

---

### Official Review · Reviewer_yJWQ · 2024-11-03

**Soundness:** 3
**Presentation:** 2
**Contribution:** 2
**Rating:** 6
**Confidence:** 5

**Summary:**

The paper applies the same framework as in Oko et al. (a paper on convergence rates of diffusion models as timesteps and/or sample size goes to infinity), to Flow Matching. Due to the application to a different model, some of the proofs are different but the results are of the same strength.

**Strengths:**

It's interesting to know that FM have the same standard guarantees as DMs.

**Weaknesses:**

Mathematically, the paper does not shine as to novelty, it mostly chains known estimates, and applies them to FM.
Like in similar papers for other models, some of the setups look like toy models, this may be because the mathematical theory is unavailable in general.

**Questions:**

1) In Theorem 1, what is a good bound for R_0?
2) Still in theorem 1, why do you chose $\sigma_{[\tau]}$ in that form? In what applications does it appear that way?
3) Can you test the sharpness of the bounds of theorem 9 for some FM famous use cases?

at line 310 "in general we generally consider" may be rephrased

---

> ### Author Response · Authors · 2024-11-19
> **Authors' replies**
>
> Thank you for your comments.  We write our replies to them.
>
> (1) Novelty:  As detailed in our response to Reviewer nHx4 (W1), the dependence of convergence rates on the choice of $\sigma_t$ and $m_t$, and the bound around the final time point of ODE are significantly different from the existing literature and novel to the best of our knowledge.
>
> (2) The setup covers various FM models, including the most popular OT-CFM (Tong et al 2023).  Note that, as in the reply to Q3 of yJWQ, the way of constructing a coupling does not affect the analysis in the paper.   We would greatly appreciate it if you could specify which aspects appear as toy models to address your concerns better.
>
> Q1:  As discussed in line 446, the constant $R_0$ can be taken so that $R_0 \geq (s+1)/min(\kappa,\bar{\kappa})$.  However, we cannot know $s$ in many practical cases, so it is not easy to set $R_0$ as this minimum value.
>
> Q2:  $\sigma_{[\tau]}=1-\tau$ ($\kappa = 1$) is the most popular choice of FM, and $\sigma_{[\tau]}=(1-\tau)^{1/2}$ ($\kappa = 1/2$) is the diffusion path (Lipman 2023, e.g.).  The choice in Theorem 1 covers practically popular ones.  If my answer does not reply to your question, please ask further.
>
> Q3: For $\kappa = 1/2$, the upper bound of Theorem 9 is of the same rate (up to an arbitrary positive $\delta$) as the lower bound of Wasserstein distance in the minimax sense, as shown in Proposition 2.  So, the bound is tight for $\kappa = 1/2$.  For other values of $\kappa$, there are no theoretical arguments for the tightness of the obtained convergence rate.  As discussed in (2) above, the result holds for a wide range of FM, such as OT-CFM.
>
> Typo:  Thank you for pointing it out.  We have fixed it in the revision.

---

> > ### Comment · Reviewer_yJWQ · 2024-11-25
> > **mainly about above points (1) and (2) + change of mind on numerical rating**
> >
> > Thank you for the reply, and for replying to my questions.
> >
> > About point (2)
> >
> > ----------------
> > Apology: About "toy model" I realize now that it was not clear what I meant, I am truly sorry for not having been specific before. I thought it would be clear that this statement has to do with the hypotheses of your main theorem, and I was expecting a discussion of these hypotheses/assumptions, but this was maybe clear only to me and not to an outside reader of my comment, sorry about that. Also, as I said, most likely no mathematical tools are available to improve things. This is also so for the diffusion version of your results (i.e. with previous work).
> >
> > *Ok so now to the concrete question.* To show that your assumptions are not over-indulgent (not toy-model-like), it would be great if you discuss to what extent (A1) - (A5) are verified by actual datasets. In particular, the regularity assumption (A1) seems hard to believe to be realistic, and I'd be happy to be proven wrong. The paper has no experiments, so the way you would prove that is by citing other papers that do have regularity verifications, but I don't know of such papers to be honest. That's why, for the time being, I stand by my previous sentence:
> >
> > "Like in similar papers for other models, some of the setups look like toy models, this may be because the mathematical theory is unavailable in general."
> >
> > Did Tong et al. had your Besov assumption, or did they have something stronger? I have missed that, but if you can point to where they have an equal or stronger assumption I'd be interested to know. But even if Tong et al. did have stronger assumptions than you do, and even if that paper is "popular", this still looks like a toy model until it is compared to actual data.
> >
> > *Note that toy models are good for our intuition. Saying "toy" makes us think of children and simple stuff, and this makes it seem dismissive, which was not my intention. I nevertheless think this is a weakness of the paper.*
> >
> >
> > About point (1): I see that actually you have some differences to Oko et al., but the paper's techniques can't help but feel incremental compared to previous work.
> >
> > Anyway, even though your replies did not change much my mind about the textual content of my evaluation, I now realize that my initial "numerical grading" (grade 5) of the paper was perhaps too severe, I think rather than "below acceptance threshold" it should go "above acceptance threshold". This is mainly because you are right that people care about these results, it gives them something to cite regarding flow matching

---

> > > ### Author Response · Authors · 2024-11-29
> > > **Thank you for your clarification**
> > >
> > > We sincerely apologize for misunderstanding your initial comments and greatly appreciate your detailed clarification. Your insights have allowed us to better address your concerns, particularly regarding the assumptions (A1)-(A5). Below, we provide a detailed discussion of these assumptions and their role in our theoretical framework. We are also willing to include these points in the final version of our paper if reviewers deem it beneficial.
> > >
> > > ### **1. Assumptions (A3) and (A4):**
> > >
> > > These assumptions concern the algorithm parameters $\sigma_t$ and $m_t$, which are specified by the user.  They are satisfied by many standard FM methods.
> > >
> > > ### **2. Assumptions (A1), (A2), and (A5):**
> > >
> > > These assumptions reflect the properties of the target probability, which is unknown in usual settings.  While we acknowledge that verifying these assumptions for specific datasets is infeasible, The primary purpose of the current and many other theoretical work is to compare the potential ability of estimators or learning methods by the worst-case analysis over a function class, revealing the dependence on important parameters such as dimensionality and smoothness degree. . These rates provide a comparative understanding of the estimator's performance.
> > >
> > > For instance, as discussed in Section 3.1, KDE with a Gaussian kernel achieves a minimax rate of $O(n^{-4/(4+d)})$, while using an optimal kernel can yield a better rate of $O(n^{-2s/(2s+d)})$ for the densities on $[0,1]^d$ with smoothness $s$. This comparison informs practical choices by highlighting the importance of kernel selection based on expected smoothness. Similarly, our work demonstrates that FM methods attain the almost minimax optimal convergence rate, which is comparable to DM methods, offering key theoretical insights into FM’s ability.
> > >
> > > ### **3. Theoretical Role of (A1)**
> > >
> > > We understand your concern regarding the practicality of  (A1). While it may not always align with real-world data, it is critical for ensuring smoothness conditions that allow rigorous analysis. Relaxing this assumption is an important direction for future work. Nonetheless, (A1) does not necessarily impose overly unrealistic conditions; for example, functions with smoothness $s$ but not $s+1$ may still be differentiable almost everywhere (e.g., ReLU is non-differentiable only at the origin). This ensures that the function class considered under (A1) is both theoretically rich and practically relevant.
> > >
> > > ### **4. Besov space**
> > >
> > > To address your question regarding Besov spaces, Tong et al. (2023) did not use such spaces because their analysis did not involve convergence rates in large-sample asymptotics. In contrast, our work focuses on deriving minimax rates, which inherently depend on the smoothness of the target density. Besov spaces, despite their complexity, have been widely recognized as effective tools for formalizing smoothness. They have been extensively used in theoretical studies on approximation and estimation accuracy  (DeVore et al., 1992; Donoho & Johnstone, 1995), and we adopt this tradition to characterize FM methods rigorously.
> > >
> > > ### References:
> > >
> > > R. A. DeVore, B. Jawerth and B. J. Lucier. (1992) Image compression through wavelet transform coding. *IEEE Transactions on Information Theory,* 38 (2) 719-746.
> > >
> > > Donoho, D. L., & Johnstone, I. M. (1995). Adapting to Unknown Smoothness via Wavelet Shrinkage. *Journal of the American Statistical Association*, *90* (432), 1200–1224.

---

### Official Review · Reviewer_nHx4 · 2024-11-04

**Soundness:** 4
**Presentation:** 3
**Contribution:** 3
**Rating:** 6
**Confidence:** 3

**Summary:**

The paper provides estimates for the 2-Wasserstein distance for the sample-based distribution obtained in the Flow-Matching framework relative to the exact distribution. These estimates depend on the number of samples used in training, the smoothness of the true distribution as an element of the Besov space, the asymptote of the growth of the conditional map at the initial time instant.
The paper considers the early stopping mode of the ODE, when the solution stops at time $T_0<1$, and the estimates of $T_0$ are also given.

**Strengths:**

- The paper is the first to present estimates for the Flow Matching framework, which shows that almost optimal minimax converges rates are achieved under several assumptions.
- The paper is well written

**Weaknesses:**

This paper contains many points in common with the paper cited therein [1].  In particular, using Besov space for target density, B-splines for its approximation, etc. Many estimates are based on thouse from [1], see, for example, Appendix A.4--A.5 of the presented paper, where the citation on [1] are explicit. In paper [1] diffusion models are considered, but as shown in paper [2], Flow Matching approach includes, under certain conditions, Diffusion models approach. Thus, generalization or obtaining similar results for Flow Matching is rather straightforward. Namely, in essence, the difference is to use Alekseev-Gröbner Theorem (Lemma 16 about error of a perturbed solution of ODE) instead of Girsanov’s Theorem (Proposition D.1 of [1] for error of a perturbed solution of SDE).
One of the main differences is the presence in the estimates of the degree of growth of the parameter $\sigma_t$ at 1, but the authors come to the well-known (empirical) conclusion that the optimal asymptotics is $\sqrt t$. Does this provide the first theoretical justification for this empirically observed optimal scaling?  How can one intuitively realize that the degree of $\sigma_t$ growth near the time point $t=1$ is important if the ODE solution is considered on the interval $[0, T_0]$, where $T_0<1$?


[1] Kazusato Oko, Shunta Akiyama, and Taiji Suzuki. Diffusion models are minimax optimal distribution estimators. volume 202, pages 26517–26582. PMLR, 4 2023

[2] Aaron Lipman, Ricky T Q Chen, Heli Ben-Hamu, Maximilian Nickel, and Matthew Le. Flow matching for generative modeling. In The Eleventh International Conference on Learning Representations, 2023

**Questions:**

- How do estimates change if you take a distribution other than the Gaussian distribution as the initial distribution $P_{[0]}$?

- Can the obtained estimates be easily extended to the case of estimation error in the total variation (TV) distance?

- Would your estimates change if you use different heuristics for Flow Matching, such as OT-minibatch?

---

> ### Author Response · Authors · 2024-11-19
> **Authors' replies**
>
> Thank you for your helpful comments. We reply to your comments.
>
> W1: Overlap with Oko et al 2023.
>
> While we acknowledge some overlap in mathematical techniques with Oko et al. 2023, there are significant difference also as listed below.
>
> (1) We have revealed the role of parameters $\sigma_t$ and $m_t$ on the upper bound of the convergence rates. No such results have been derived in the literature for FM and DM.  To obtain the results, the mathematical techniques for the proof of main theorems (Appendix C) require significant refinements from Oko et al 2023.
>
> (2) As detailed in the global comments, the generalization analysis of DM and FM have significant difference.  Beyond this difference, you might notice that the bound $W_2(P_0, P_{T_0})$ requires completely novel arguments using the uniform Lipschitz constant in Lemma 10, which are based on the form of $\sigma_t$ and $m_t$ considered in this paper and the assumption (A5).
>
> (3) Appendices A4 and A5 summarize known results (e.g., B-splines and Gaussian integral bounds) and do not intend novelty. We have explicitly stated this for clarity in the revision.
>
> (4) We do not think $\sigma_t = \sqrt{t}$ is an empirically popular choice for FM unlike DM.  In fact, the most popular choice of path construction for FM is $x_t = (1-t)x_1 + t x_0$, which corresponds to $\sigma_t = t$ and $m_t = 1-t$ ($\kappa = 1$).
>
> W2: degree of growth of $\sigma_t$
>
> Section 3.1 discusses how $\sigma_t$ acts as a smoothing parameter, analogous to the bandwidth in kernel density estimation.  The early stopping time $T_0$ depends on $n$ ($T_0 = n^{-R_0}$ in the paper), and the growth rate $\kappa$ in $\sigma_t=t^\kappa$ controls the smoothing parameter at the stopping time $t=T_0$.  It is well known that the smoothing parameter of a nonparametric estimator essentially affects on the convergence rate.  This is why $\kappa$ is important for the convergence rate and the performance of the FM estimator.
>
> Q1: other than Gaussian distribution
>
> This is an interesting question, and in fact we are working on this extension.  In the current proof, special properties of Gaussian distribution help us making bounds at many places.  Mathematically, it is challenging to extend it to general distributions.
>
> Q2: Extension to TV
>
> As discussed in Global Responses, it is not straightforward to extend the (almost) minimax optimal convergence rate to the TV distance.  This also illustrates the significant difference between the ODE and SDE for analyzing the generalization bounds.
>
> Q3: Heuristics for FM such as OT-minibatch.
>
> The current analysis does not depend on the **joint** distribution of the source $N(0,I_d)$ and the target $P_{true}$.  Thus, for any algorithms to match samples in the two distributions, including OT-minibatch, the theoretical results hold.  We have mentioned it at line 158 in the revision .

---

> > ### Comment · Reviewer_nHx4 · 2024-12-03
> >
> > I thank the authors for the answers. I have read the responses to the other reviewers and the updated version of the article. I keep my score 6.

---

### Author Response · Authors · 2024-11-19
**Global Responses**

We thank all the reviewers for their constructive feedback. Below, we address the specific questions and concerns raised. We appreciate your insights and look forward to further discussions to improve our manuscript.  We have uploaded a revision, in which we have addressed your comment.  Some major changes in the revised manuscript are highlighted in blue for clarity.

Several reviewers have raised concerns regarding the novelty of our generalization analysis compared to that of Diffusion Models (DM). As a global response, we would like to clarify the relationship between the generalization analysis of Flow Matching (FM) and DM.

While it is true that the probability dynamics of the reverse diffusion model can be represented via an ODE, specifically the probability flow ODE (Song et al., 2021), this does not imply equivalence in the generalization analysis of DM and FM.

The vector field for the probability flow ODE is expressed as:
$$
v_t(x_t)=a_t x_t + b_t \nabla \log p_t(x_t).
$$
Here, $a_t$ and $b_t$ are constants related to $\sigma_t$ and $m_t$ (or $\beta_t$ in the standard DM parameterization). In DM, only the score function $\nabla \log p_t(x_t)$ is estimated, and its generalization analysis primarily evaluates the accuracy of this score estimator.  In contrast, for FM, the generalization analysis involves evaluating the accuracy of the entire vector field. As evident from the equation above, the accuracy of the score estimation is insufficient to ensure the accuracy of the vector field.

Furthermore, in DM or SDE-based frameworks, bounds on KL divergence and Total Variation (TV) distance can be derived from score function estimates via Girsanov’s theorem. However, to the best of our knowledge, no analogous KL bound exists for ODEs in terms of the vector field. This highlights a fundamental distinction between the generalization analysis for ODEs and SDEs.

---

### Author Response · Authors · 2024-11-24
**Reviewers' responses would be appreciated**

Dear Reviewers,

We submitted our responses to your review comments several days ago and provided detailed comments on your questions and concerns.  We would appreciate it if you could give us any further feedback.

Thank you,
Authors

---

### Meta-Review · Area_Chair_haGL · 2024-12-12

**Metareview:**

The paper shows flow matching, or ODE-based generative models, achieves almost minimax optimal convergence, which complements the prior literature (e.g., Oko et al) on the minimax optimality of SDE-based generative models. Given the significance of flow matching, this result is a welcomed addition to the growing literature, and will be of interest to the ICLR audience.

**Additional Comments On Reviewer Discussion:**

During the rebuttal, it was clarified how the results here are different from Oko et al, e.g., using the Alekseev-Grobner Theorem instead of the Girsanov’s Theorem.

---

### Decision · Program_Chairs · 2025-01-22

Accept (Poster)